# Understanding the role of depth in the neural tangent kernel for overparameterized neural networks

## Abstract

Overparameterized fully-connected neural networks have been shown to behave like kernel models when trained with gradient descent, under mild conditions on the width, the learning rate, and the parameter initialization. In the limit of infinitely large widths and small learning rate, the kernel that is obtained allows to represent the output of the learned model with a closed-form solution. This closed-form solution hinges on the invertibility of the limiting kernel, a property that often holds on real-world datasets. In this work, we analyze the sensitivity of large ReLU networks to increasing depths by characterizing the corresponding limiting kernel. Our theoretical results demonstrate that the normalized limiting kernel approaches the matrix of ones. In contrast, they show the corresponding closed-form solution approaches a fixed limit on the sphere. We empirically evaluate the order of magnitude in network depth required to observe this convergent behavior, and we describe the essential properties that enable the generalization of our results to other kernels.

## 1 Introduction

Machine learning approaches have demonstrated a remarkable ability in helping solve a multitude of problems across a large span of tasks. Whether the task is classification, prediction, or image generation, some variants of the combination of a neural network plus gradient descent have managed to achieve superhuman ability in some instances (LeCun et al., 1998; Silver et al., 2018; Vaswani et al., 2017). In recent years, a particular observation has been made regarding neural networks that are overparameterized. While previous common thought indicated that these large networks would fall prey to overfitting, this conclusion is being challenged empirically (Belkin, 2021). This phenomenon is not quite well understood and it has led to works analyzing the learning dynamics of overparameterized models updated with gradient descent (Liu et al., 2020; 2022; Jacot et al., 2018a). While these works provide insight as to the learning dynamics of fully-connected neural networks that are overparameterized, Jacot et al. (2018a), in particular, offer a closed-form solution to the gradient flow based on a kernel that is recursively computed. This result offers the possibility of approximately predicting the output of an overparameterized neural network learned through gradient descent, without explicitly training the model. This comes at the cost of computing a kernel over a particular dataset, which involves the computation of expectations.

We focus on fully-connected ReLU networks, which allows us to both speed up the computation of the kernel and obtain a more interpretable closed-form solution. Leveraging this more interpretable closed-form solution, we study the role of depth in the limiting kernel of infinitely wide fully-connected ReLU networks. Our contributions address two central aspects of the effect of increasing depths: 1) the convergence of the kernel, established in Proposition 4, and 2) the limiting solution to the output of a fully-connected ReLU network under infinitely wide hidden layers and infinitely small learning rate via Theorem 3. Our results apply to arbitrary data with support on the sphere and, in contrast to previous literature, do not require any assumptions on the spectrum of the Hermite expansion or

the Mercer decomposition of the kernel. The limiting solution we obtain does not require any non-invertibility assumption as in Xiao et al. (2020) (see Theorem 3 and the detailed discussion following it). While the kernel of Proposition 4 does converge to a constant matrix, the limiting solution mentioned in 2) converges to a well-defined limit. We achieve this key result by making use of rough differential equations machinery. In contrast to Hanin & Nica (2020), we study the deterministic limit of the neural tangent kernel when the width is much larger than the depth: while we allow the depth to increase to infinity, the rate at which it does so is much slower than the widths of the hidden layers of a neural network. We offer a summary of results in relation to ours in table 1 in Appendix E.

## 2 Related Work

The learning dynamics of overparameterized neural networks have been extensively studied through the lenses of kernel methods and Hessian-based analysis. A key development in this area is the neural tangent kernel (NTK), introduced by Jacot et al. (2018a), which describes how infinitely wide fully-connected neural networks trained with gradient descent evolve linearly in function space. The NTK framework formalizes how, under common assumptions—particularly Gaussian initialization and wide-layer limits—neural networks behave similarly to kernel methods during training. Arora et al. (2019b) extend the NTK to convolutional neural networks (CNN), further highlighting the framework's capacity to capture learning dynamics.

Subsequent work has reinforced the kernel-based interpretation of training dynamics through analyses of the Hessian matrix. Liu et al. (2020; 2022) and Belkin (2021) demonstrate that the loss landscape of overparameterized neural networks often exhibits near-linearity, with low-curvature regions and small-norm Hessians, supporting the NTK-based approximation. These findings suggest that network outputs are relatively stable during training, especially for wide architectures with standard initialization. Lee et al. (2020) provide an in-depth empirical analysis of NTK models and their performance compared to finite-width neural networks of various architectures (e.g. fully-connected, CNNs). One key observation is that NTKs often outperform finite-width networks, yet are usually surpassed by conventional CNNs.

The NTK typically requires Monte Carlo estimation of expectations over Gaussian distributions, especially when nonlinearities from activations are involved. This reliance on sampling can be computationally expensive and introduces variance in the resulting kernel evaluations. However, closed-form expressions have been derived for certain activation functions, such as ReLU and leaky ReLU (Tsuchida et al., 2018). Additionally, while prior work has largely focused on width, less attention has been paid to how depth affects NTK sensitivity to initialization and the associated multiplicity in network outputs. Among the few works addressing this dimension, Bietti & Bach (2021) show that for the uniform measure on the sphere, the reproducing kernel (c.f. reproducing kernel Hilbert space or RKHS) leads to the same representation power regardless of the network depth. This raises the question of the significance of depth for the NTK. Lee et al. (2022) present an analysis of deep narrow MLPs and CNNs as depth goes to infinity. They provide a trainability guarantee, given the right initialization, by showing convergence to a limiting kernel and a vanishing training error. However, this initialization is not close to common initialization methods used in practice, and it is still unclear whether deep narrow networks possess better generalization capabilities than infinitely wide networks. Further insights are provided by Nguyen et al. (2021), who derive an asymptotic lower bound on the smallest eigenvalue of the NTK through the Hermite expansion of the kernel. This result can be used to derive bounds on the generalization of the model (Arora et al., 2019a) and gives a better grasp on understanding the role that depth plays in both convergence and generalization. Murray et al. (2023) characterize the full spectrum of the NTK via the Hermite expansion for arbitrary datasets on the sphere. They recover an empirical observation that the eigenvalues of the NTK follow power law decay with respect to the size of the training set; Li et al. (2024) extend this line of results to general domains beyond the sphere. In addition to the previous result from Jacot et al. (2018a) regarding the convergence to kernel regression in the infinite width case, they also establish a uniform convergence bound to the NTK regressor for the output of the trained

model. This improves upon the pointwise convergence bounds found in Lee et al. (2019); Arora et al. (2019b); Allen-Zhu et al. (2019). As in the initial paper by Jacot et al. (2018a), the limiting kernel is observed to be deterministic. This is contrasted with the work of Hanin & Nica (2020), where the authors characterize the NTK directly through the ratio of its second and first moment. They show that if the ratio of depth to width grows large, the NTK has a much higher variance than mean, implying that it is highly stochastic. Xiao et al. (2020) characterize three different phases describing the behaviour of the limiting kernel of infinitely wide neural networks. They show that the "ordered phase" and the "critical phase" can lead to generalization when trainable, while the "chaotic phase" collapses the mean-predictor to a (data-independent) constant predictor exponentially fast. Their proof in the ordered phase relies on the invertibility of a partciular matrix and does not generalize to cases where this matrix is singular. This observation for the ordered phase has also been observed by Seleznova & Kutyniok (2022), who show that the NTK of infinitely large ReLU networks converges to a singular matrix. This reinforces the fact that small changes in the limiting kernels (as depth increases) can lead to significant changes in the mean-predictor.

## 3    NOTATION

To highlight the role of the depth $L$ of a neural network, we denote elements that depend on layer $l \in \{1, \ldots, L\}$ with a superscript $(l)$, e.g. $\kappa^{(l)}$. To emphasize the special role of the last layer in the analysis, the superscript $(L)$ is sometimes used in mathematical objects to remind the reader of its link to a neural network of depth $L$, e.g. $\kappa^{(L)}$ refers to a network of depth $L$ while it can also stand on its own as a mathematical object. The width of a layer $l$ is denoted by $n_l$, with $n_0$ referring to the input dimension. Dependence on a time parameter $t$ and the size $n$ of a dataset is sometimes indicated with a subscript to emphasize their importance, but is otherwise omitted. A dataset of size $n$ is denoted $X$, understood as an $n \times n_0$ matrix, where each row $i$ is written as $x_i^\top$ (i.e. the $x_i$'s are column vectors). It is assumed that all rows are different. Activation functions are denoted by $\sigma$, while uppercase $\Sigma$ is reserved for computing "covariances" (see Definition 1). The limiting deterministic kernels of Jacot et al. (2018a) are represented using $\Theta_\infty^{(L)}$, and $\bar{\Theta}_\infty^{(L)}$ for their normalized version (see Definition 4); the notation $\kappa$ is used when referring to general kernels and $\bar{\kappa}$ for the normalized version. For the sake of simplicity, we consider neural networks with one-dimensional outputs (i.e., $n_L = 1$). Therefore, kernels refer in this context to functions $\mathbb{R}^{n_0} \times \mathbb{R}^{n_0} \to \mathbb{R}_+$. We also write $\Theta(A)$ for the component-wise application of a kernel $\Theta$ to the entries of a matrix $A$. Specifically, $\kappa(XX^\top)$ denotes applying the kernel to all pairwise dot products in $X$, where $XX^\top$ is the matrix containing those dot products. Similarly, for any function $g : \mathbb{R} \to \mathbb{R}$, the entry-wise application to a matrix $A$ is denoted by $g(A)$. The notation $A \leftharpoonup_{i,j} A'$ refers to the matrix obtained by replacing column $i$ of matrix $A$ with column $j$ of matrix $A'$. The vector of ones of length $n$ is denoted $\mathbf{1}_n$. Finally, the sphere of dimension $n_0 - 1$ is denoted $S^{n_0-1}$.

## 4    BACKGROUND ON THE NTK AND OVERPARAMETERIZATION

Jacot et al. (2018a) show that, under overparameterization and i.i.d. standard normal weight initialization, a fully-connected neural network of arbitrary depth $L$ exhibits learning dynamics that converge to those of kernel gradient flow in the infinite-width limit. They also provide a recursive formula to compute the kernel $\Theta_\infty^{(L)}$ to which gradient descent converges (see Theorems 1 and 2 from Jacot et al. (2018a)). However, evaluating $\Theta_\infty^{(L)}$ relies on computing high-dimensional expectations and can potentially also be subject to sample inefficiency in the approximation, motivating the search for a more readily computable kernel.

To make the kernel more practical, one may ask whether an efficient closed-form expression can be derived for particular activation functions. Of particular interest is the representation of $\Theta_\infty^{(L)}$ in closed-form when using ReLU activations (Tsuchida et al., 2018), due their empirical popularity and methodological appeal in theoretical analysis. In the rest of this paper, we will study this kernel for increasing $L$, with ReLU activation and $\mathcal{N}(0,1)$ initialization. To

this end, let us first introduce important definitions and recap the recursive formulation of $\Theta_\infty^{(L)}$ by Jacot et al. (2018a).

**Definition 1** ((mean) Covariance of neurons $\Sigma^{(l)}$). *Let $x$ and $x'$ be two inputs in $\mathbb{R}^{n_0}$. The covariances of neurons from inputs $x$ and $x'$ at each layer $l$ are defined recursively as*

$$\Sigma^{(1)}(x, x') := \frac{1}{n_0} x^\top x', \quad \Sigma^{(l+1)}(x, x') := \mathbb{E}_{f \sim \mathcal{N}(0, \Sigma^{(l)})}[\sigma(f(x))\sigma(f(x'))]$$

*where $f \sim \mathcal{N}(0, \Sigma^{(l)})$ is an infinite vector indexed through the notation $f(x)$ and $f(x')$ and each vector $(f(x), f(x'))^\top \sim \mathcal{N}\left(0, \Sigma^{(l)}(x, x')\right)$. We also define the variant of $\Sigma^{(l)}$ where we replace $\sigma$ with its derivative $\dot\sigma$:*

$$\dot\Sigma^{(l+1)}(x, x') := \mathbb{E}_{f \sim \mathcal{N}(0, \Sigma^{(l)})}[\dot\sigma(f(x))\dot\sigma(f(x'))].$$

**Definition 2** (Neural tangent kernel (NTK)). *For inputs $x$ and $x'$, the neural tangent kernel of the neural network $f(\cdot; \theta)$ with parameters $\theta \in \mathbb{R}^P$ is given by*

$$\Theta^{(L)}(x, x') = \sum_{p=1}^{P} \frac{\partial f(x; \theta_p)}{\partial \theta_p} \otimes \frac{\partial f(x'; \theta_p)}{\partial \theta_p}.$$

**Theorem 1** (Jacot et al. (2018a)). *Suppose we have a fully-connected neural network of depth $L$ with non-linear activation. In the limit as layer widths $n_1, \ldots, n_{L-1} \to \infty$, the neural tangent kernel (see Definition 2) $\Theta^{(L)}$ converges in probability to a deterministic limiting kernel:*

$$\Theta^{(L)} \to \Theta_\infty^{(L)} \otimes I_{n_L},$$

*where $\Theta_\infty^{(l)}$ is defined recursively by*

$$\Theta_\infty^{(1)}(x, x') := \Sigma^{(1)}(x, x')$$
$$\Theta_\infty^{(l+1)}(x, x') := \dot\Sigma^{(l+1)}(x, x')\Theta_\infty^{(l)}(x, x') + \Sigma^{(l+1)}(x, x').$$

We remark that, although we assume $n_L = 1$ for the sake of simplicity, Theorem 1 is stated in its general form for any output dimension $n_L \in \mathbb{N}$. We also note that this is the version of the theorem without biases ($\beta = 0$ in the context of Jacot et al. (2018a)). This theorem is key in the convergence results obtained in the next section (Proposition 4 and Theorem 2). We are now ready to state the simplified formula for positively correlated inputs (Proposition 1).

**Proposition 1.** *For ReLU activation and perfectly positively correlated inputs $x$ and $x'$, i.e. $\rho = 1$, it holds that*

$$\Sigma^{(L)}(x, x') = \frac{1}{n_0 2^{L-1}} \|x\|_2 \|x'\|_2, \quad \dot\Sigma^{(L)}(x, x') = \frac{1}{2}$$

*and*

$$\Theta_\infty^{(L+1)}(x, x') = \frac{1}{2}\Theta_\infty^{(L)}(x, x') + \frac{1}{n_0 2^L} \|x\|_2 \|x'\|_2 = \frac{L+1}{n_0 2^L} \|x\|_2 \|x'\|_2.$$

*Proof sketch.* Note that $\frac{x^\top x'}{\|x\| \|x'\|} = 1$ and the product $\sigma^2(f(z))$, where $z \sim \mathcal{N}(0, 1)$, follows a squared rectified gaussian distribution, and that $\mu = 0$ implies $x^\top x' \geq 0$ with probability $\frac{1}{2}$. $\qquad\square$

**Definition 3** (Correlation coefficient of $\Sigma^{(L)}(x, x')$). *The correlations of neurons from inputs $x$ and $x'$ are defined as*

$$\rho^{(L)}(x, x') := \frac{\Sigma^{(L)}(x, x')}{\sqrt{\Sigma^{(L)}(x, x)\Sigma^{(L)}(x', x')}}.$$

*Note that $\rho^{(L)}(x, x') \in [-1, 1]$.*

**Proposition 2** (Arora et al. (2019b)[1]). *For datapoints $x$ and $x'$ with $\rho \in [-1, 1[$, it holds that*

$$\rho^{(L+1)}(x, x') = \frac{\sqrt{1 - (\rho^{(L)}(x, x'))^2}}{\pi} + \frac{\rho^{(L)}(x, x') \arcsin \rho^{(L)}(x, x')}{\pi} + \frac{1}{2}\rho^{(L)}(x, x')$$

$$\dot{\Sigma}^{(L+1)}(x, x') = \frac{\arcsin \rho^{(L)}(x, x')}{2\pi} + \frac{1}{4}$$

*and*

$$\Theta_\infty^{(L)}(x, x') = \|x\|_2 \|x'\|_2 \Theta_\infty^{(L)} \left( \frac{x}{\|x\|_2}, \frac{x'}{\|x'\|_2} \right)$$

*for a fully-connected neural network with ReLU activation.*

With these results, we achieved our goal of obtaining a closed-form expression for the $\Theta_\infty^{(L)}$ corresponding to an overparametrized (infinite-width), fully-connected ReLU network with no biases. This, in turn, allow us to aim at characterizing the output of such neural network, as done in the rest of the section. Indeed, from Proposition 2 and Proposition 2 from Jacot et al. (2018b), we can immediately observe a few facts regarding the input data:

case a) If all datapoints lie on the unit sphere $S^{n_0-1}$, the NTK is invertible for $L \geq 2$ (Proposition 2 from Jacot et al. (2018b)).

case b) If all datapoints are pairwise not colinear, i.e. $x_i^\top x_j < \|x_i\|_2 \|x_j\|_2$ for $i \neq j$, then the NTK is invertible (for $L \geq 2$) since we can project them to different points on the sphere through the canonical projection.

case c) If we map points from $\mathbb{R}^{n_0}$ to the sphere $S^{n_0}$ by embedding them in a space of dimension $n_0 + 1$ and projecting them with the inverse stereographic projection (see Definition 7), the embedding of the datapoints satisfies $x_i^\top x_j = 1$ for all $x_i, x_j$ in the dataset.

If one of these cases holds, the following proposition provides a closed-form expression for the approximation of the output of a fully-connected neural network.

**Proposition 3** (Jacot et al. (2018a)). *Let $X$ be a dataset of size $n$ (with entries $x_i^\top$) and let $f^*$ and $f_0$ respectively refer to the learned function and the neural network after the initialization. If the limiting kernel $\kappa = \Theta_\infty^{(L)} \left( XX^\top \right)$ is invertible, the output of the neural network converges to*

$$f_\infty(x) = f_0(x) + \kappa_x^\top \kappa^{-1}(y^* - y_0),$$

*where*

$$\kappa_x = \Theta_\infty^{(L)} \left( xX^\top \right), \qquad (y^*)_i = f^*(x_i), \qquad (y_0)_i = f_0(x_i), \qquad i = 1, \ldots, n$$

*as time $t \to \infty$, i.e. the number of gradient descent updates increases.*

Motivated by Proposition 3 and the requirement to have an invertible $\kappa$, we identify **two** regimes of generalization: datapoints can lie on either a **1)** non-compact manifold (i.e. $\mathbb{R}^{n_0}$) or **2)** a compact manifold (i.e. $S^{n_0-1}$). The compact regime results in a simplifying assumption for the analysis that follows in this section. Note that one can project any dataset without any pair of colinear datapoints in $\mathbb{R}^{n_0}$ on $S^{n_0-1}$ using the canonical projection. The kernel $\kappa$ will thus be invertible. If colinear points exist, an inverse stereographic projection embedding on $S^{n_0}$ will result in an invertible $\kappa$.[2]

## 5 Limiting Kernel as Depth Increases

While in the previous section, the depth $L$ is fixed and the width goes to infinity, no mention is made of the effect of increasing both the depth and the width. Such insights into the

---

[1]See also Cho & Saul (2009) for the complete derivation.

[2]In the context of learning the parameters of a neural network, we assume that one first projects onto the sphere and then fixes the projected data during the training phase.

additional effect of depth would provide a tangible frontier for the representation power of fully-connected neural networks and their generalization capabilities. In this section, we describe how the term $\kappa_x$ from Proposition 3 approaches a fixed limit for each $x$ as $L \to \infty$. The NTK will also approach this limit when $L \to \infty$, with $L \in o(\min_{l=1,\ldots,L-1} n_l)$. Note that this setting is different from Hanin & Nica (2020), where the ratio of depth to width can be arbitrary; interestingly, when the depth grows faster than the width, there is no convergence to a deterministic limit for the NTK and it is stochastic. In this section, we always assume ReLU activation for $\Theta_\infty^{(L)}$ and $\Sigma^{(L)}$. For a concise list that summarizes the assumptions made in this section, we refer the reader to Appendix A.

The following lemma demonstrates that $\rho$ converges to 1 for each pair of datapoints as $L$ goes to infinity. This result is a key ingredient in the propositions and theorems that follow.

**Lemma 1** (Convergence of $\rho^{(L)}$). *If $\rho^{(1)}(x, x') \in ]-1, 1[$, then $\rho^{(L)}(x, x') \to 1$ as $L \to \infty$.*

In the equation of Proposition 3, the terms $\kappa_x$ and $\kappa$ can both be normalized by a scalar and the resulting vector-matrix product is left unchanged. Specifically, if $\Theta_\infty^{(L)}$ is normalized such that its diagonal elements are all equal to 1, some immediate results follow from Propositions 1 and 2. These results are shown in Proposition 4 and Theorem 2.

**Definition 4** (Normalization of the $\Theta_\infty^{(L)}$ kernel). *For $x, x' \in S^{n_0 - 1}$, the normalized version of $\Theta_\infty^{(L)}$ is defined by*

$$\bar{\Theta}_\infty^{(L)}(x, x') = \frac{n_0 2^{L-1} \Theta_\infty^{(L)}(x, x')}{L}.$$

**Definition 5.** *We define the function $h : [-1, 1] \to \mathbb{R}$ as*

$$h(z) = \frac{z \arcsin(z)}{\pi} + \frac{\sqrt{1 - z^2}}{\pi} + \frac{z}{2}.$$

Using the definitions above, we state the Proposition 4 and the Theorem 2. The proofs can be found in Appendix C.

**Proposition 4** (Alternative formulation of $\bar{\Theta}_\infty^{(L)}$). *The equality*

$$\bar{\Theta}_\infty^{(L+1)}(x, x') = \frac{L}{L+1} h'\left(\rho^{(L)}(x, x')\right) \bar{\Theta}_\infty^{(L)}(x, x') + \frac{1}{L+1} h\left(\rho^{(L)}(x, x')\right)$$

*holds $\forall x, x' \in S^{n_0 - 1}$. Moreover, the values in the normalized kernel are all found in the interval $[0, 1]$.*

**Theorem 2** (Convergence of $\bar{\Theta}_\infty^{(L)}$). *For any $x, x' \in S^{n_0 - 1}$, the value $\bar{\Theta}_\infty^{(L)}(x, x')$ strictly increases to 1 as $L \to \infty$.*

The result above can be taken to be a major obstacle to the analysis of $\Theta_\infty^{(L)}\left(x^\top X^\top\right)\left(\Theta_\infty^{(L)}\left(XX^\top\right)\right)^{-1}$ since the positive determinant of $\Theta_\infty^{(L)}$ converges to 0. However, Theorem 3, which is one of our key contributions, demonstrates that for a fixed $x$, the term $\Theta_\infty^{(L)}\left(x^\top X^\top\right)\left(\Theta_\infty^{(L)}\left(XX^\top\right)\right)^{-1}$ converges to some limit as $L$ increases to infinity. The proof of this theorem requires a function defined in Definition 6 and whose key properties are provided as Proposition 5. The required background on rough differential equations is provided in Appendix D.

**Definition 6.** *We define the function $\psi_d$ for $d \in \mathbb{R}^+$ as*

$$\psi_d(z) = \begin{cases} \frac{1}{1 + \exp\left(\frac{-2z}{d(1 - z^2)}\right)} & \text{if } z \in ]-1, 1[ \\ 1 & \text{if } z = 1 \\ 0 & \text{if } z = -1. \end{cases}$$

**Proposition 5.** *The function $\psi_d$ has the following key properties on $[-1, 1]$:*

$$\psi_d(-1) = 0 \tag{1}$$

$$\psi_d(1) = 1 \tag{2}$$

$$\psi_d \in \mathcal{C}^\infty \tag{3}$$

$$\lim_{d \to 0^+} \frac{\frac{d^k}{dz^k}\psi_d(z)}{d^j} = 0 \qquad \forall j, k \in \mathbb{N}_0. \tag{4}$$

**Theorem 3** (Rough differential equation (RDE) solution). *In the compact regime, for each dataset $X$ of size $n$ (with entries $x_i^\top$) and $x \in S^{n_0-1}$, there exists a sequence of paths $v_{ij}^{(L)} : [0, 1] \to \mathbb{R}^n$ such that*

$$\lim_{L \to \infty} v_{ij}^{(L)}(t) = 0 \quad \forall t \in [0, 1]$$

*and the rough path lift $\boldsymbol{v}^{(L)} : \Delta_{0,1} \to \mathbb{R}^{n \times n+1}$ of $v_{ij}^{(L)}$ with $p = 1$ **drives** the solution $\mathbf{u}^{(L)}$ of a differential equation*

$$\frac{d}{dt}u_i^{(L)}(t) = 0 \quad \forall i \in \{1, \dots, n\},$$

*and whose projection $\left(\mathbf{u}^{(L)}\right)^1 = u^{(L)}$ onto 1-tensors satisfies the equality*

$$u_i^{(L)}(1) = \bar{\Theta}_\infty^{(L)} \left(x^\top X\right)^\top \left(\bar{\Theta}_\infty^{(L)} \left(X^\top X\right)\right)^{-1}.$$

*Here, $\Delta_{0,1}$ refers to the set $\{(s, t) : 0 \leq s \leq t\}$. Specifically,*

$$\bar{\Theta}_\infty^{(L)} \left(x^\top X\right) \left(\bar{\Theta}_\infty \left(X^\top X\right)\right)^{-1} < C(x)\mathbf{1}_n^\top$$

$$\left\| \left(\bar{\Theta} \left(X^\top X\right)\right)^{-1} \bar{\Theta} \left(X^\top x\right) \right\|_2 \in \mathcal{O}(n)$$

*for $L$ large enough. Moreover, when $x$ is free and $n$ is fixed, the function $C$ is continuous and hence bounded on $S^{n_0-1}$.*

*Proof.* We define the matrix $A_n^{(L+1)}(t)$ with

$$A_n^{(L+1)}(t) = \bar{\Theta}_\infty^{(L)} \left(XX^\top\right) + \psi_\mathcal{D}(2t - 1) \left(\bar{\Theta}_\infty^{(L+1)} \left(XX^\top\right) - \bar{\Theta}_\infty^{(L)} \left(XX^\top\right)\right)$$

$$\mathcal{D} = \det\left(\bar{\Theta}_\infty^{(L+1)} \left(XX^\top\right)\right) \det\left(\bar{\Theta}_\infty^{(L)} \left(XX^\top\right)\right)$$

for $t \in [0, 1]$ and dataset $X = \{x_i\}_{i=1}^n$ of size $n$. We also define $b_n^{(L+1)}(t)$ with

$$b_n^{(L+1)}(t) = \bar{\Theta}_\infty^{(L+1)} \left(x^\top X^T\right).$$

From the system $A_n^{(L+1)}(t)u(t) = b_n^{(L+1)}(t)$, we take the derivative with respect to $t$ and obtain the system

$$\left(\frac{d}{dt}A_n^{(L+1)}(t)\right) u(t) + A_n^{(L+1)}(t) \left(\frac{d}{dt}u(t)\right) = \frac{d}{dt}b_n^{(L+1)}(t).$$

Note that the solution $u(t)$ depends implicitly on $n, L$ and $x$. This will be made obvious later in the proof, but it is hidden for cleaner notation. By Cramer's rule, the solution to the system is

$$u'(t)_i = \frac{\sum_j \det\left(A_n^{(L+1)}(t) \hookleftarrow_{i,j} Z_A\right)}{\det\left(A_n^{(L+1)}(t)\right)} + \frac{\det\left(A_n^{(L+1)}(t) \hookleftarrow_{i,1} Z_b\right)}{\det\left(A_n^{(L+1)}(t)\right)}, \tag{5}$$

where $Z_A, Z_b$ are defined by

$$Z_A = -\left(\frac{d}{dt}A_n^{(L+1)}(t)\right) \text{diag}(u(t)), \quad Z_b = \frac{d}{dt}b_n^{(L+1)}(t) = \mathbf{0}_n,$$

where the boldface $\mathbf{0}_n$ denotes the vector of 0's of length $n$. By property (4) of $\psi_{\mathcal{D}}$, we obtain the sequence of inequalities

$$\frac{\det\left(A_n^{(L+1)}(t) \hookleftarrow_{i,j} \frac{d}{dt} A_n^{(L+1)}(t)\right)}{\det\left(A_n^{(L+1)}(t)\right)} \tag{$v_{(i,j)}$}$$

$$\leq \frac{\det\left(A_n^{(L+1)}(t) \hookleftarrow_{i,j} \frac{d}{dt} A_n^{(L+1)}(t)\right)}{\det\left(\bar{\Theta}_\infty^{(L+1)}\left(XX^\top\right)\right)^{\psi_{\mathcal{D}}(2t-1)} \det\left(\bar{\Theta}_\infty^{(L)}\left(XX^\top\right)\right)^{1-\psi_{\mathcal{D}}(2t-1)}}$$

$$\leq \frac{\det\left(A_n^{(L+1)}(t) \hookleftarrow_{i,j} \frac{d}{dt} A_n^{(L+1)}(t)\right)}{\det\left(\bar{\Theta}_\infty^{(L+1)}\left(XX^\top\right)\right) \det\left(\bar{\Theta}_\infty^{(L)}\left(XX^\top\right)\right)} \to 0 \quad \text{as } L \to \infty,$$

for $L$ large enough. Note that we obtain the last inequality above through the fact that for $L$ large enough, the strictly positive determinants are all smaller than 1. In addition, because the function $\psi_{\mathcal{D}}$ is infinitely smooth, the terms $v_{(i,j)}$ are all of bounded total variation (see Definition 11 with $p = 1$ in the appendix). For the same reason and using (4), we have that the $v_{(i,j)}$ converge to 0 in the 1-variation metric. By Lyons Universal Limit theorem (Lyons, 1998) from rough path theory (see Definition 12 in the appendix), the solution $u^{(L+1)}(t)$ (where we make the dependence on $L + 1$ explicit) converges to the solution $u_\infty(t)$ that solves the system $u'_\infty(t) = \mathbf{0}_n$. Hence, $u_\infty(t)_i$ is a constant dependent on $i$ and $x$. We have a limiting solution $u_\infty(t)$ that is bounded for each $x$. Because the Itô-Lyons map $\Phi$ (Definition 14 in appendix) is continuous and locally Lipschitz in any $p$-norm variation topology, and because the rough path lift of $v_{(i,j)}$ is continuous, the entire solution is bounded on the compact set $S^{n_0-1}$, i.e. there is a bound $C' < \infty$ such that $u_\infty(t) < C'\mathbf{1}_n$ for all $x$ (note that $u_\infty(t)$ depends on $x$ but not $C'$). $\qquad\square$

In summary, Theorem 3 provides a limiting solution for the expression $\bar{\Theta}_\infty^{(L)}\left(x^\top X^\top\right) \left(\bar{\Theta}_\infty^{(L)}\left(XX^\top\right)\right)^{-1}$ by solving systems of equations. The continuous solutions interpolate between solutions for different values of $L$. These systems are differentiated and formulated as rough differential equations that converge to a limiting RDE. The solution to this RDE at $t = 1$, i.e. the limiting expression, is dependent on $x$ and non-trivial. Moreover, when evaluated at $x_i \in X$, $i \in \{1, \ldots, n\}$, the limit is $e_i$, the $i^{\text{th}}$ standard basis vector. While $\bar{\Theta}_\infty^{(L)}\left(XX^\top\right)$ approaches a (constant) singular matrix, our proof shows that the limiting expression is well-defined. This distinguishes our approach from that of Xiao et al. (2020) (see Appendix D.3 in Xiao et al. (2020)), where $\bar{\Theta}_\infty^{(L)}\left(XX^\top\right)$ is decoupled as a (constant) data-independent matrix and a data-dependent matrix whose limit is assumed to be invertible. Given that for any dataset $X$, Theorem 2 guarantees that $\bar{\Theta}_\infty^{(L)}\left(XX^\top\right)$ converges to 1, the proof in Xiao et al. (2020) would not apply.

An immediate consequence of this result is that if the depth $L \in o(\min_{1 \leq l < L-1} n_l)$ and the width $(\min_{1 \leq l \leq L_1} n_l)$ both go to infinity (i.e. the ratio of depth to width going to 0), the kernel of the output $f_\infty$ of the neural network reaches a limiting expression. This limiting expression characterizes the effect of depth on infinitely wide fully-connected ReLU networks whose inputs lie on the sphere $S^{n_0-1}$. For ReLU networks, it is possible to easily extend this result to the non-compact regime (i.e. general domain $\mathbb{R}^{n_0}$). By Proposition 2, there is a closed-form to $\Theta_\infty$ for general data points in $\mathbb{R}^{n_0}$. In the statement of the proposition, the canonical projection on the sphere is provided, but a similar result is obtained for a stereographic projection.

## 6 Experiments and Theoretical Implications

In the proof of Theorem 3 from the previous section, we can identify the key properties used to derive the results, in order to distill the essence of the type of kernels that lead to similar limiting behaviour. We summarize these properties for data on $S^{n_0-1}$.

**Arbitrary sequence of kernels $\kappa^{(L)}$ satisfying the requirements of the theorem**

- $\kappa^{(L)}(x, x) \geq \kappa^{(L)}(x_1, x_2)$ for any $x, x_1, x_2 \in S^{n_0 - 1}$ and all $L \in \mathbb{N}$.

- There is some $\hat{L} \in \mathbb{N}$ such that $\kappa^{(L)}\left(XX^\top\right)$ is positive definite for any $X = \left\{x_i \mid x_i \in S^{n_0-1}, i = 1, \ldots, n\right\}$ of size $n$ and $L \geq \hat{L}$.

- $\lim_{L \to \infty} \det\left(\bar{\kappa}^{(L)}\left(XX^\top\right)\right) = 0$ for data in $S^{n_0 - 1}$.

By inspection of the definition of $\rho^{(L)}$, it can be observed that it satisfies the criteria of the list above. Another example is given by the sequence $\eta^{(L)}$ of kernels that are defined recursively by $\eta^{(L+1)}(x, x') = h(\kappa^{(L)}(x, x'))$ and $\eta^{(1)}(x, x') = x^\top x'$, where $h(z) = (1 + e^{-z})^{-2}$ (see Proposition 7 in the appendix).

In order to better understand the theoretical insights from the previous section, we empirically evaluate the convergence rates of $\bar{\Theta}_\infty^{(L)}$, $\rho^{(L)}$ and $\eta$ as $L$ increases. We illustrate this convergent behavior in figure 1, where we generate a dataset $X$ and a point $x$ from the uniform distribution ($n_0 = 128$) and we canonically project them to the sphere. We then plot the evolution of the values for depths $L = 1, \ldots, 30$. Note that this depth limit is sufficient to show convergence. Each curve in the plots corresponds to a different pair of inputs (either from $X$ or between $x$ and $X$; e.g. $\bar{\Theta}_\infty^{(L)}(x_i, x_j)$). It is immediate at first glance that both $\rho^{(L)}$ and $\eta^{(L)}$ converge to a fixed value rather quickly as the depth increases. In contrast, from the plot of $\bar{\Theta}_\infty^{(L)}\left(XX^\top\right)$, while it is seemingly the case that off-diagonal values converge to some value strictly smaller than 1, Theorem 2 demonstrates that they converge to 1. In addition to the synthetic dataset $X$, we perform the same experiment on the MNIST dataset (LeCun et al., 2010), whose tensors are converted to vectors that are normalized to lie on the sphere. We report the results in figure 3 in Appendix F. In both datasets, the convergence rate is sublinear for $\bar{\Theta}_\infty^{(L)}\left(XX^\top\right)$ and it is possible to observe this directly in the proof of the theorem: $K$ is taken to be much larger than $L$, and given an approximation threshold of $0 < \delta \ll 1$, the exponential approximation to $(1 - \delta)^{K+1}$ requires that $K\delta \gg 0$, i.e. $K$ is much larger than $\frac{1}{\delta}$. The convergence rate to 1 can in fact be shown to be logarithmic. Nevertheless, by inspection of the proof of Theorem 3, we can see that $\left(v_{(i,j)}\right)$ converges to 0 exponentially faster than $\det\left(\bar{\Theta}_\infty^{(L)}\left(XX^\top\right)\right)$. This serves to show the convergence to the limiting solution is fast, provided the determinant is small. We numerically evaluate this determinant by computing it in figure 2 for MNIST in Appendix F. We hypothesize that small determinants indicate fast convergence to the limiting solution. This would suggest that depths $L$ that are not very large are sufficient for a good approximation to the limiting solution. While we do not explore any other kernels experimentally, we refer the reader to the list of criteria in this section to derive any other candidate to explore its convergence properties.

Note that Bietti & Bach (2021) and Li et al. (2024) tell us that the representation power of $\Theta_\infty^{(L)}$ does not change as $L \to \infty$. However, if we apply our limiting result to the mean-field regime of Chizat & Bach (2018), we find that each particle approaches the deterministic limit by inspection of Proposition 3 and Theorem 3. It is therefore possible to analyze the many-particle limit of very wide and deep fully-connected neural networks since these are well approximated by $f_\tau$ for a proper stopping time $\tau$ (Li et al., 2024). In addition, the proof technique of Theorem 3 can be adapted to other kernels that arise from other architectures such as CNNs.

## 7 CONCLUSION

In this article, we provide a detailed analysis of behaviour of the deterministic kernel $\Theta_\infty^{(L)}$ as $L \to \infty$. We observe that under the conditions of Jacot et al. (2018a) and with $L \in o(\min_{l=1,\ldots,L-1} n_l)$, a fully-connected ReLU neural network approaches a limiting solution for any $x$ given a fixed dataset $X$. The studied kernel exhibits behaviour consistent with the ordered phase. In contrast to Xiao et al. (2020), we do not assume that the limiting

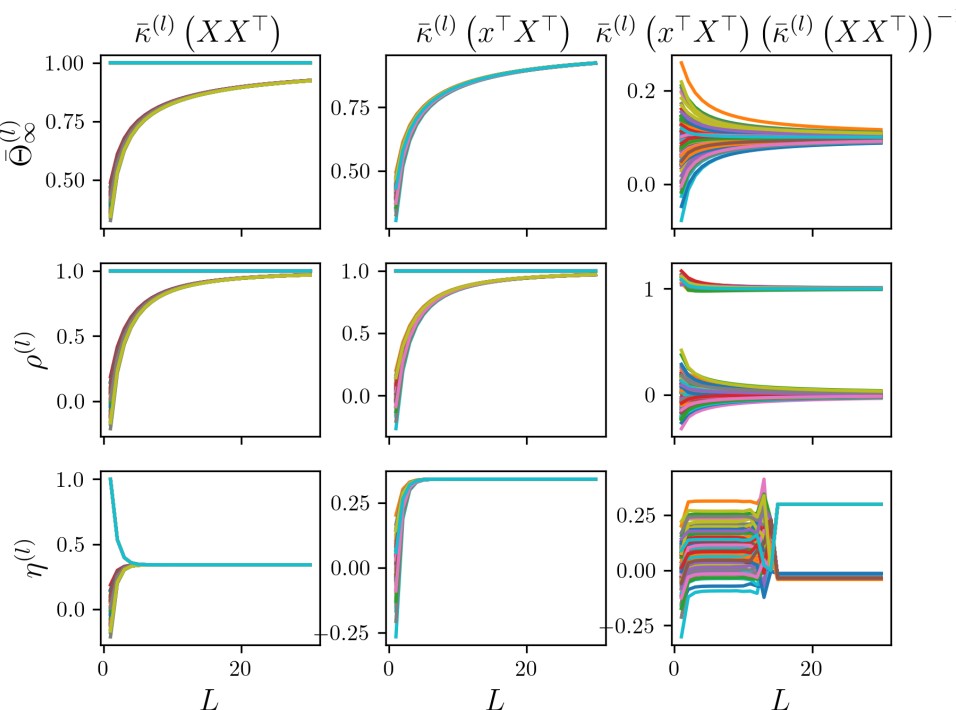

Figure 1: Convergence rate of $\kappa$ on $X$ and point $x$. We model 3 different expressions dependent on an arbitrary $\kappa$ in each column. The particular choice of $\kappa$ is given by each row (see label on the left).

kernel can be written as the sum of a constant and a non-singular matrix. We generalize the existence of a limiting solution for $\kappa_x \kappa^{-1}$ to include the singular case. While our results are for data in $S^{n_0-1}$, we can extend this result to the general domain as long as long as there is no colinearity, or by taking the stereographic projection in a space with one additional dimension. On the one hand, we have shown the convergence rates of a non-exhaustive list of kernels as depth $L$ increases. For the limiting kernels $\kappa^{(L)} = \Theta_\infty^{(L)}$, we have observed that the convergence is extremely slow and would require very large $L$ before reaching the limit. On the other hand, small depths $L$ are required to approximate the limit of $\kappa_x \kappa^{-1}$. Specifically, we demonstrate empirically that, while convergence for the limiting kernel is sublinear, the convergence for the limiting kernel is experimentally fast. We hypothesize that this will be the case in many scenarios. Studying the settings in which this is true is left as a future research direction. Finally, we provided a list of key properties that were necessary to obtain our results to generalize to other kernels.

We believe that our work can help researchers to better understand the role of depth in the deterministic limiting kernels of overparameterized neural networks. Future research should envisage to better understand the behaviour of kernels that arise in the context of other architectures such as CNNs and architectures with skip connections. As mentioned, the setting of Hanin & Nica (2020) is outside the purview of our analysis as the stochasticity of the NTK becomes highly relevant when $L \gg \max_{l=1,\ldots,L-1} n_l$. Nevertheless, by using the analytical tools of rough differential equations, we raise the hypothesis that there might exist a "pointwise" limit to the NTK when $L \to \infty$, where it is implied that the convergence is with respect to each sample path.

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
