## A  Theoretical assumptions

For the proofs in Section 5, we make the following list of explicit assumptions:

**Activation** $\sigma$: ReLU activation.

**Output dimension**: The output dimension considered is $n_L = 1$ for a network of depth $L$.

**Biases**: The neural networks do not contain any biases (i.e. $\beta = 0$).

**Data**: The data has a fixed representation on $S^{n_0-1} \subseteq \mathbb{R}^{n_0}$ and no duplicates are in $X$.

**Kernel** $\Theta_\infty^{(L)}$: Limiting kernel of a fully-connected ReLU neural network of depth $L$, output dimension $n_L = 1$ and without biases, under infinite width (i.e. $\min_{1 \leq l \leq L-1} n_l \rightarrow \infty$).

Introducing biases via $\beta > 0$ changes $\Sigma^{(L)}$ and $\Theta_\infty^{(L)}$. Therefore, Proposition 1 does not hold in its exact form. However, we can show that in the limit $L \rightarrow \infty$, the kernel $\bar{\Theta}_\infty^{(L)}$ converges to a constant matrix whose entries are all equal. By the same token, we can apply the ideas of Theorem 3, and we obtain a limiting expression for $f_\infty$. In the general list of properties for $\kappa^{(L)}$ that is found in Section 6, the value of $\hat{L} = 2$ as the limiting kernel becomes positive definite on the sphere for $L \geq 2$.

## B  Inverse stereographic embedding and further definitions

**Definition 7** (Inverse stereographic projection embedding). *Let $m \in \mathbb{N}$, $q \in S^m$, and $\pi_{proj} : S^m \rightarrow \mathbb{R}^{m+1}$ be the stereographic projection from $S^m$ to $\mathbb{R}^{m+1}$ through the point $q$. The point $q$ can be thought as the "point at infinity" in $\mathbb{R}^{m+1}$. The **inverse stereographic projection embedding** $\pi_{inv} : \mathbb{R}^m \rightarrow S^m$ is defined as*

$$\pi_{inv} = \pi_{proj}^{-1} \circ \pi_{emb},$$

*where $\pi_{emb} : \mathbb{R}^m \rightarrow \mathbb{R}^{m+1}$ is defined by*

$$\pi_{emb}(x) = (x_1, \ldots, x_m, 0).$$

**Definition 8** (Logarithmic convergence). *Given a sequence $\{x_n\}_{n=1}^\infty$, the sequence has a logarithmic convergence rate if $\lim_{n \rightarrow \infty} x_n = x$ for some $x \in \mathbb{R}$ and*

$$\lim_{n \rightarrow \infty} \frac{|x_{n+1} - x|}{|x_n - x|} = 1, \quad \lim_{n \rightarrow \infty} \frac{|x_{n+2} - x_{n+1}|}{|x_{n+1} - x_n|} = 1.$$

## C  Additional Lemmas, Propositions, and Theorems

**Lemma 1** (Convergence of $\rho^{(L)}$). *If $\rho^{(1)}(x, x') \in \,]-1, 1[$, then $\rho^{(L)}(x, x') \rightarrow 1$ as $L \rightarrow \infty$.*

*Proof.* We observe that $\rho^{(L+1)}(x, x') = h(\rho^{(L)})$, where $h$ is the function

$$h(z) = \frac{z \arcsin(z)}{\pi} + \frac{\sqrt{1 - z^2}}{\pi} + \frac{z}{2}.$$

This function has derivative $h'(z) < 1$ on any fixed interval $[a, b] \subsetneq [-1, 1]$ such that $b \neq 1$ and is continuously differentiable on the same interval. Therefore, if $H_n(z) = (h \circ \cdots \circ h)(z)$ denotes the $n^{\text{th}}$ power composition of $h$, we have $H_L(z) \rightarrow \beta$, some unique fixed-point, uniformly on $]-1, 1[$. This can be proved by observing that the domain of $H_L$ is a compact set and that for any metric $d$, the distance $d(H_L(z_1), H_L(z_2)) < d(z_1, z_2)$ for $z_1 \neq z_2$ (first show that $d(H_L(z), z)$ is continuous and has a minimum which is 0). Furthermore, we have that $h(z) \geq z$ on $[-1, 1]$, with strict inequality on $[-1, 1[$. This implies that $\beta = 1$ and the proof is finished. $\qquad \square$

The following proposition is the early-stopping variant of proposition 3. We include a proof sketch for the reader, although we wish to underscore that this result is already known in the literature.

**Proposition 6** (closed-form for $f_\tau$; section 5 from Jacot et al. (2018a)). *Given $\tau < \infty$ and the spectrum $\Lambda(\kappa)$, the output $f_\tau$ is given by*

$$f_\tau(x) = f_0(x) + \kappa_x^\top \kappa^{-1} \bar{\Lambda}(y_0 - y^*),$$

*where $\kappa_x, y_0, y^*$ are as in proposition 3, and*

$$\bar{\Lambda}_{ij} = \begin{cases} \exp(-\lambda_i \tau) - 1 & \text{if } i = j \\ 0 & \text{otherwise} \end{cases}$$

*is the diagonal matrix applying the function $g_\tau(\lambda) = \exp(-\lambda\tau) - 1$ elementwise to the elements of $\Lambda(\kappa)$.*

*Proof sketch.* From section 5 in Jacot et al. (2018a), the output $f_t(x)$ of the neural network is given by

$$f_t(x) = f_\infty(x) + \alpha_0^{(f_0 - f^*)}(x) + \sum_{i=1}^n \exp(-\lambda_i t)\alpha_i^{(f_0 - f^*)}(x)$$

where the terms $\alpha_i^{(f_0 - f^*)}$ are the component functions of the eigenvalue decomposition of $f_0(x) - f^*(x)$. Notice that we can write

$$\begin{aligned} f_\infty(x) + \alpha_0^{(f_0 - f^*)}(x) &= f_0(x) - \sum_{i=1}^n \alpha_i^{(f_0 - f^*)}(x) \\ &= f_0(x) - \kappa_x^\top \kappa^{-1}(y_0 - y^*), \end{aligned}$$

where the second equality follows from $\alpha_i^{(f_0 - f^*)}(x) = \kappa^{-1}(y_0 - y^*)_i e_i$ for standard basis vectors $e_i \in \mathbb{R}^n$. This implies the following equality:

$$f_t(x) = f_0(x) + \kappa_x^\top \kappa^{-1} \bar{\Lambda}(y_0 - y^*).$$

We have an expression for the output $f_\tau(x)$ for any input $x$ and stopping time $\tau$. $\square$

The following proposition provides a proof sketch of the convergence of $\eta^{(L)}$ as $L \to \infty$ and other properties that satisfy the requirements for Theorem 3.

**Proposition 7** (Convergence of $\eta^{(L)}$). *The values $\eta^{(L)}(x, x')$ converge to a unique $\beta > 0$ for all $x, x' \in S^{n_0 - 1}$ as $L \to \infty$. Moreover, the kernels $\eta^{(L)}$ are positive definite on $S^{n_0 - 1}$ and satisfy $\eta^{(L)}(x, x) \geq \eta^{(L)}(x_1, x_2)$.*

*Proof sketch.* As $L \to \infty$, all values converge to the same limit since the derivative of $h$ is strictly smaller than 1 on $[-1, 1]$. The kernels $\eta^{(L)}$ also satisfy $\eta^{(L)}(x, x) \geq \kappa^{(L)}(x_1, x_2)$ since $h(z)$ is monotone increasing in $z \in [-1, 1]$. The kernels are all positive definite on $S^{n_0 - 1}$ since the function $h$ is analytic on $[-1, 1]$ and it has infinitely many even and odd terms in its power series expansion at 0 (i.e. not and even or odd function) that are strictly positive (Gneiting, 2013). $\square$

The following is the proof of Proposition 4.

**Proposition 4** (Alternative formulation of $\bar{\Theta}_\infty^{(L)}$). *The equality*

$$\bar{\Theta}_\infty^{(L+1)}(x, x') = \frac{L}{L+1} h'\left(\rho^{(L)}(x, x')\right) \bar{\Theta}_\infty^{(L)}(x, x') + \frac{1}{L+1} h\left(\rho^{(L)}(x, x')\right)$$

*holds $\forall x, x' \in S^{n_0 - 1}$. Moreover, the values in the normalized kernel are all found in the interval $[0, 1]$.*

*Proof.* The fact that the values are all contained in $[0,1]$ is immediate from the definition of $\bar{\Theta}_\infty^{(L)}$ and Proposition 2. Now,

$$\Theta_\infty^{(L+1)}(x,x') = \dot{\Sigma}^{(L+1)}(x,x')\Theta_\infty^{(L)}(x,x') + \Sigma^{(L+1)}(x,x')$$

$$= \frac{1}{2}h'\left(\rho^{(L)}(x,x')\right)\Theta_\infty^{(L)}(x,x')$$

$$+ \frac{1}{n_0 2^L}h\left(\rho^{(L)}(x,x')\right),$$

where the first equality comes from Theorem 1 and the second equality uses Propositions 1 and 2. This implies

$$\bar{\Theta}_\infty^{(L+1)}(x,x') = \frac{Ln_0 2^L}{(L+1)n_0 2^{L-1}}$$

$$\times \frac{n_0 2^{L-1}\bar{\Theta}_\infty^{(L)}(x,x')}{L}\frac{1}{2}h'\left(\rho^{(L)}(x,x')\right)$$

$$+ \frac{1}{L+1}h\left(\rho^{(L)}(x,x')\right)$$

$$= \frac{L}{L+1}h'\left(\rho^{(L)}(x,x')\right)\bar{\Theta}_\infty^{(L)}(x,x')$$

$$+ \frac{1}{L+1}h\left(\rho^{(L)}(x,x')\right),$$

where the equalities come from the definition of $\Theta_\infty^{(L+1)}(x,x')$ and the normalization factors of $\bar{\Theta}_\infty^{(L+1)}$ and $\bar{\Theta}_\infty^{(L)}$. $\qquad\square$

The following contains the proof of Theorem 2.

**Theorem 2** (Convergence of $\bar{\Theta}_\infty^{(L)}$). *For any $x,x' \in S^{n_0-1}$, the value $\bar{\Theta}_\infty^{(L)}(x,x')$ strictly increases to $1$ as $L \to \infty$.*

*Proof.* We have a system describing $\bar{\Theta}_\infty^{(L+1)}$,

$$\begin{pmatrix} \bar{\Theta}_\infty^{(L+1)} \\ 1 \end{pmatrix} = \begin{pmatrix} \frac{L}{L+1}h'\left(\rho^{(L)}(x,x')\right) & \frac{1}{L+1}h\left(\rho^{(L)}(x,x')\right) \\ 0 & 1 \end{pmatrix}\begin{pmatrix} \bar{\Theta}_\infty^{(L)}(x,x') \\ 1 \end{pmatrix}$$

where we define the $2 \times 2$ matrix on the right-hand side to be $A^{(L)}$. For now, fix $L$ and observe the product

$$A^{(L+K)}\cdots A^{(L)},$$

which we define to be the $2 \times 2$ matrix $A^{[L:L+K]}$. First off, we can show that

$$A_{11}^{[L:L+K]} = \frac{L}{L+K+1}\prod_{k=0}^{K}h'\left(\rho^{(L+k)}(x,x')\right) \to 0$$

as $K \to \infty$ (recall we fixed $L$). The same convergence would be true if $L \in o(K)$ as $L,K \to \infty$. We also obtain a sequence of inequalities bounding $A_{12}^{[L:L+K]}$, where

$$A_{12}^{[L:L+K]} = \begin{pmatrix} A_{11}^{[L+2:L+K]} & A_{12}^{[L+2:L+K]} \end{pmatrix}\begin{pmatrix} \frac{1}{L+2}h'\left(\rho^{(L+1)}(x,x')\right)h\left(\rho^{(L)}(x,x')\right) + \frac{1}{L+2}h\left(\rho^{(L+1)}(x,x')\right) \\ 1 \end{pmatrix}$$

$$= \frac{1}{L+K+1}\sum_{k=0}^{K}h\left(\rho^{(L+k)}(x,x')\right)\prod_{k'=k+1}^{K}h'\left(\rho^{(L+k')}(x,x')\right)$$

$$\geq \frac{1-\delta}{L+K+1}\sum_{k=0}^{K}(1-\delta)^{K-k}$$

$$= \frac{1-\delta}{L+K+1}\sum_{k=0}^{K}(1-\delta)^{k}$$

$$= \frac{1-\delta}{L+K+1}\frac{1-(1-\delta)^{K+1}}{\delta}$$

if $L$ is large enough such that $\min\left(h'\left(\rho^{(L)}(x,x')\right), h\left(\rho^{(L)}(x,x')\right)\right) \geq 1 - \delta$ for a $\delta \in \,]0,1[$. The second equality was obtained by repeatedly doing the matrix-vector products using matrices $A^{(L)}, \ldots, A^{(L+K)}$. We observe that by the identity $(1+x)^\alpha \approx e^{\alpha x}$ for $x$ small and $\alpha x$ large, for $\delta$ small enough

$$\frac{1-\delta}{L+K+1}\frac{1-(1-\delta)^{K+1}}{\delta} \approx \frac{1-\delta}{L+K+1}\frac{1-e^{-K\delta}}{\delta}$$

$$\approx (1-\delta)\frac{K}{L+K+1}$$

which, once we unfreeze $L$, goes to $1 - \delta$ when $L \in o(K)$ as $L \to \infty$. Since $L$ is allowed to become arbitrarily large, we can take $\delta$ arbitrarily small by Lemma 1 (note that $K$ is still large enough such that $K\delta \gg 0$). $\qquad\square$

## D  ROUGH PATH THEORY

In this section, we provide some background notions on rough path theory. For more information, see Lyons (1998).

**Definition 9** (Truncated tensor algebra $T^{(m)}$). *The truncated tensor algebra $T^{(m)}$ of $\mathbb{R}^d$ is given by*

$$T^{(m)}\left(\mathbb{R}^d\right) = \bigoplus_{i=0}^{m}\left(\mathbb{R}^d\right)^{\otimes i},$$

*where $\left(\mathbb{R}^d\right)^{\otimes 0} \cong \mathbb{R}^d$ and $\bigoplus$ is the direct sum operator.*

**Definition 10** (Projective norm $\pi$ on tensor powers). *Given any norm $\|\cdot\|$ on $\mathbb{R}^d$, it can be used to define the projective norm $\pi$ on any tensor product $\left(\mathbb{R}^d\right)^{\otimes m}$ by*

$$\pi(x) = \inf\left\{\sum_{i=1}^{m}\|a_i\|\|b_i\| : x = \sum_{i=1}^{m} a_i \otimes b_i\right\}.$$

**Definition 11** ($p$-variation metric). *Let $\Delta_{0,1}$ be the simplex $\{(s,t) : 0 \leq s \leq t \leq 1\}$ and $p \geq 1$. Let $\mathbf{X}, \mathbf{Y}$ be continuous maps $\Delta_{0,1} \to T^{(\lfloor p \rfloor)}\left(R^d\right)$ and let $\mathbf{X}^j$ (resp. $\mathbf{Y}^j$) denote the projection of $\mathbf{X}$ (resp. $\mathbf{Y}$) onto its $j$-tensor component. The $p$-variation metric $d_p$ is defined by*

$$d_p\left(\mathbf{X},\mathbf{Y}\right) = \max_{j=1,\ldots,\lfloor p \rfloor}\sup_{0=t_0<t_1<\cdots<t_m=1}\left(\sum_{i=0}^{m-1}\|\mathbf{X}^j - \mathbf{Y}^j\|^{\frac{p}{j}}\right)^{\frac{j}{p}},$$

*where the supremum is taken over all finite partitions $\{0 = t_0 < t_1 < \cdots < t_m = 1\}$ of $[0,1]$.*

**Definition 12** (Rough path (Lyons, 1998)). *A continuous function $\mathbf{X} : \Delta_{0,1} \to T^{(\lfloor p \rfloor)}\left(R^d\right)$ is a $p$-geometric rough path if there exists a sequence of paths with finite 1-variation (i.e. bounded variation) $X(1), X(2), \ldots$ such that*

$$\mathbf{X}(l)_{s,t} = \left(1, \int_{s<s_1<t} dX(l)_{s_1}, \ldots,\right.$$

$$\left.\int_{s<s_1<\cdots<s_{\lfloor p \rfloor}<t} dX(l)_{s_1} \otimes \cdots \otimes dX(l)_{\lfloor p \rfloor}\right)$$

*converges in the $p$-variation metric to $\mathbf{X}$ as $l \to \infty$. The map from $X(l)$ to $\mathbf{X}(l)$ is called the **rough path lift** of $X(l)$.*

**Definition 13** (Rough differential equation). *Let $V_{i,j}$ for $i = 1, \ldots, e$ and $j = 1, \ldots, d$ be functions that have at least $\lfloor p \rfloor$ bounded derivatives and the $\lfloor p \rfloor$-th derivatives are $\alpha$-Hölder continuous for $\alpha > p - \lfloor p \rfloor$. A rough differential equation takes the form*

$$(dY_t)_i = \sum_{j}^{d} V_{i,j}\left(Y_t\right)\left(dX_t\right)_j,$$

*where $Y$ is the solution to the differential equation, $X$ is the the driving signal, and both $X$ and $Y$ admit a rough path lift to a p-geometric rough path. If $\mathbf{Y}$ and $\mathbf{X}$ are the corresponding rough paths, we can also say that $\mathbf{Y}$ solves the differential equation driven by $\mathbf{X}$.*

**Definition 14** (Itô-Lyons map (Lyons, 1998)). *p-geometric rough paths $\Delta_{0,1} \to T^{(\lfloor p \rfloor)}\left(R^d\right)$ take value in the group $G\Omega_p\left(R^d\right)$ embedded in $T^{(\lfloor p \rfloor)}\left(R^d\right)$. Given a rough differential equation, the Itô-Lyons map is the map $\Phi : G\Omega_p\left(R^d\right) \to G\Omega_p\left(R^e\right)$ from a geometric rough path $\mathbf{X}$ to a geometric rough path $\mathbf{Y}$ solving the rough differential equation driven by $\mathbf{X}$.*

## E  RELATED WORK SUMMARY

In this section, we present a summary of our results and their relation with the available literature of Section 2 on the subject. This summary is presented in Table 1.

| Architecture | Activation | Results and relevant works |
|---|---|---|
| Wide, fixed depth, fully-connected | General | • Convergence to NTK on the sphere and general domain; Hessian is approximately zero (Jacot et al., 2018a; Belkin, 2021; Liu et al., 2020; 2022). 
 • Spectrum characterized via Hermite expansion (Nguyen et al., 2021; Murray et al., 2023; Li et al., 2024). 
 • Power law decay for NTK eigenvalues (Murray et al., 2023; Li et al., 2024). 
 • Tight bound on smallest eigenvalue of NTK (Nguyen et al., 2021). |
| Kernel model (uniform measure) | N/A | RKHS remains constant regardless of depth (Bietti & Bach, 2021). |
| Infinite width, depth, fully-connected | ReLU, general | • NTK behaves stochastically (as $\max_{1 \le l \le L-1} \frac{L}{n_l} > 0$ becomes large)(Hanin & Nica, 2020). 
 • Kernel limit as depth increases (as $\lim_{L,n_1,\ldots,n_{L-1}\to\infty} \max_{1 \le l \le L-1} \frac{L}{n_l} = 0$) **(Our work 2025)**. 
 • Results for both spherical and general domains (Hanin & Nica, 2020); **(Our work 2025)**. |

Table 1: Summary of literature review and contributions

## F  FURTHER EXPERIMENTS

In this section, we present further experiments relating to the theoretical results presented in Section 5. We report the mean squared error, accuracy and F1 score on MNIST data in Table 2. For a small dataset of size 500 and a depth of 30, we observe good performance. In Figure 3, we can see that the limiting solution $\bar{\Theta}_\infty^{(L)}\left(XX^\top\right)$ is well approximated at such depths. Therefore, we can model the output of a very wide and deep network using minimal resources. To validate that we are indeed reaching the limiting solution from Theorem 3, we plot the scale of the determinant of the limiting NTK (see discussion in Section 6) in Figure 2.

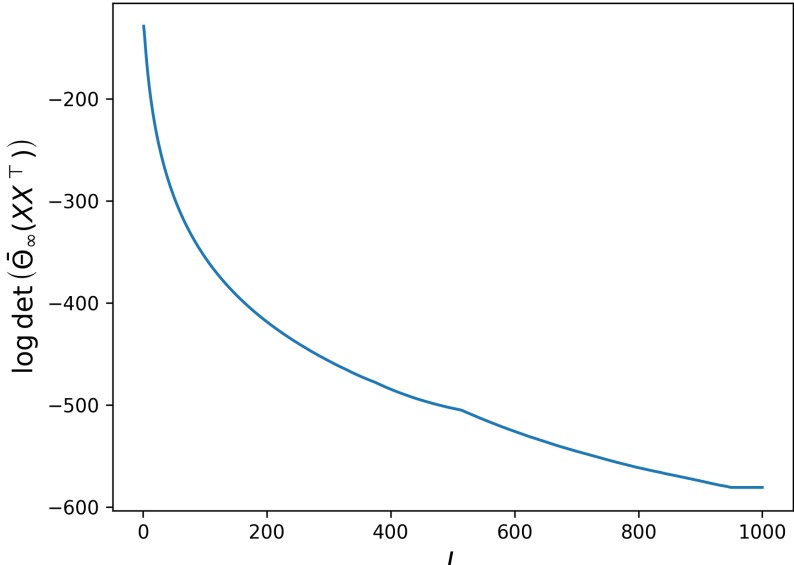

Figure 2: Evolution of the determinant of $\bar{\Theta}_\infty^{(L)}\left(XX^\top\right)$ with increasing depth $L$.

| Metric | |
| --- | --- |
| Average error (stdev) | $6.435\,(0.075)$ |
| Accuracy | $0.874$ |
| F1 score | $0.870$ |

Table 2: Various metrics on kernel regression model for MNIST. The kernel used is $\bar{\Theta}_\infty^{(L)}\left(XX^\top\right)$. The size of the dataset was set to $|X| = 500$ and the depth to $L = 30$.

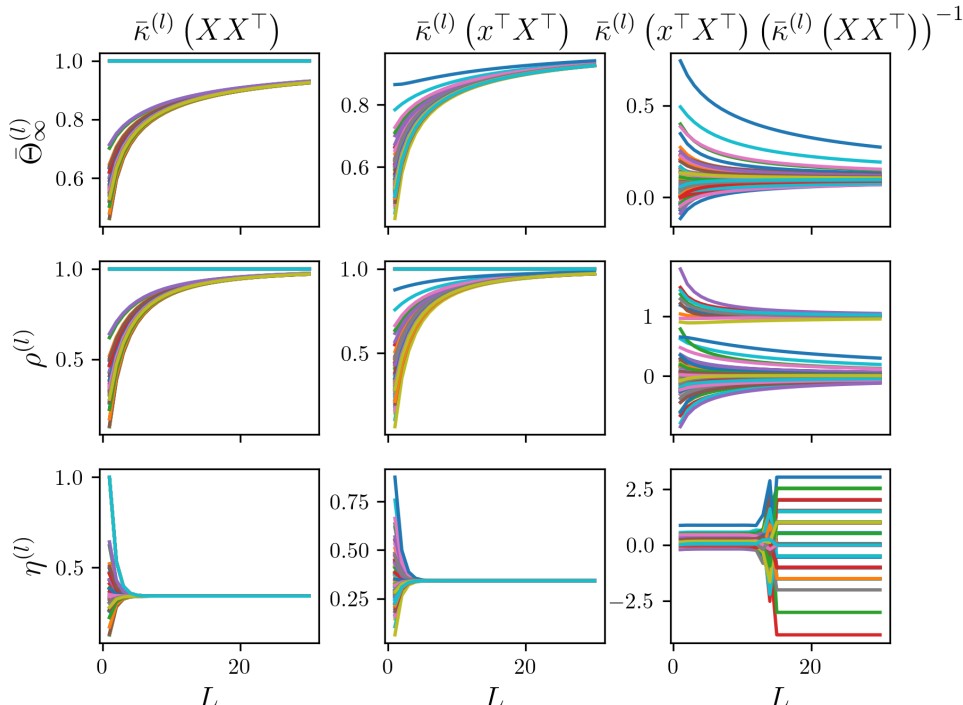

Figure 3: Convergence rate of $\kappa$ on $X$ and point $x$. We model 3 different expressions dependent on an aribtrary $\kappa$ in each column. The particular choice of $\kappa$ is given by each row (see label on the left).