# OpenReview forum: "Understanding the role of depth in the neural tangent kernel for overparameterized neural networks."
_ICLR.cc/2026/Conference — Submitted to ICLR 2026_

### Official Review · Reviewer_JL34 · 2025-10-20

**Soundness:** 1
**Presentation:** 2
**Contribution:** 1
**Rating:** 2
**Confidence:** 4

**Summary:**

This paper studies the NTK regime of neural networks in the limit of increasing depth $L \to \infty$, i.e., the regime obtained by first taking the infinite-width limit $n \to \infty$ and then the infinite-depth limit $L \to \infty$. The analysis is restricted to the *ordered phase*, where the initialization variance $W_{ij} \sim \mathcal{N}(0, 1/n)$ leads to vanishing gradients as depth increases. In this setting, the NTK $\Theta(X,X)$ approaches a singular matrix as $L \to \infty$. The authors study the behavior of the *mean predictor*, defined as $\Theta(x, X)\Theta^{-1}(X, X)$, which corresponds to the expected kernel regression solution under the NTK. The main result appears to be a proof that, despite the convergence of $\Theta(X,X)$ to a singular matrix in this vanishing-gradient regime, the mean predictor itself admits a well-defined limiting form.

**Strengths:**

- **Relevance:** In general, scaling limits of neural networks beyond the standard infinite-width regime are in important and relevant topic in deep learning theory, and the paper aims to advance this line of work.

**Weaknesses:**

- **Lack of novelty:**  The presented results have appeared in various forms in prior work. For instance, Xiao et al. (2020) [1] show the existence of a well-defined limit for the mean predictor in Eq. (16). Related results and discussions of the limitations of the "$n \to \infty$ then $L \to \infty$" limit also appear e.g. in Seleznova & Kutyniok (2022) [2] and Hayou et al. (2022) [3]. This regime has long been considered trivial, as the covariance between distinct inputs converges to one, leading to effectivelly constant predictions for any input. Moreover, it seems like the paper uses unnecessarily complex machinery of rough differential equations to prove convergence of the mean predictor, which is a result that can be obtained directly through standard linear-algebraic arguments.

- **Restricted initialization setting:**  The analysis considers only the initialization $W_{ij} \sim \mathcal{N}(0, 1/n)$, corresponding to the *ordered phase*, in which gradients vanish with depth. Prior works cited above studied a broader class of initializations and show that the behavior of the "$n \to \infty$ then $L \to \infty$" regime depends critically on this choice. The present paper does not acknowledge this and, in fact, never explicitly defines the initialization variance.

- **Lack of relevant literature discussion:**  Existing work on the regime $n \to \infty$, $L \to \infty$, $L/n \to 0$ is not discussed. The paper does not reference or contrast its results with any prior analyses of this limit.

- **Misrepresentation of prior results:**  The paper incorrectly summarizes the results of Hanin & Nica (2019) [4], which concern the *double-scaling* regime where $L, n \to \infty$ with $L/n \to \lambda > 0$. The text repeatedly claims that Hanin & Nica study the case where "the ratio of depth to width is unbounded" or where $L \gg n$, which is incorrect.

- **Lack of clarity and technical precision:** There are multiple problems with clarity and technical precision in the text. A few examples are:
  - The initialization distribution is never specified, so the source of randomness in the concentration results is unclear.
  - "Case a" (lines 221–222) states that "if data points lie on a unit sphere, the NTK is invertible." This is trivially false, with a counterexample of taking the same point on a sphere multiple times. Moreover, the text references "Proposition 2 of Jacot et al (2018)" here. However, Jacot et al (2018) [5] does not contain Preposition 2.
  - "Case c" (lines 227–228) refers to "stereographic projection from $\mathbb{R}^{n_0}$ to $\mathbb{S}^{n_0-1}$." Such a projection does not exist.

## References

[1] Xiao, L., Pennington, J., & Schoenholz, S. (2020). *Disentangling trainability and generalization in deep neural networks.* ICML.
[2] Seleznova, M., & Kutyniok, G. (2022). *Analyzing finite neural networks: Can we trust neural tangent kernel theory?* MSML.
[3] Hayou, S., Doucet, A., & Rousseau, J. (2022). *The curse of depth in kernel regime.* NeurIPS Workshop.
[4] Hanin, B., & Nica, M. (2020). *Finite depth and width corrections to the neural tangent kernel.* ICLR.
[5] Jacot, A., Gabriel, F., & Hongler, C. (2018). *Neural tangent kernel: Convergence and generalization in neural networks.* NeurIPS.

**Questions:**

- What is meant by “stereographic projection from $\mathbb{R}^{n_0}$ to $\mathbb{S}^{n_0-1}$” mentioned several times in the paper?
- Why is it necessary to rely on rough differential equations to obtain the main result?
- How do the presented results differ from or improve upon the mentioned existing work?

---

> ### Author Response · Authors · 2025-11-24
> **Reply**
>
> We would like to thank the reviewer for thoroughly engaging with our paper. The insights derived from the review proved vital in better situating our work in the literature and differentiating with existing results by emphasizing the novelty of theoretical results. We answer each concern in detail below.
>
> ##### Nota bene
> In what follows, we assume the reviewer used $n$ to refer to the width of layers instead of the size of the dataset as in the paper. We will follow this convention when replying to the reviewer's comments for consistency.

---

> ### Author Response · Authors · 2025-11-24
> **Lack of novelty**
>
> The results from Xiao et al. [2020] describe three regimes of kernel behaviours: the chaotic, critical and ordered phase. Our results align with the mean-predictor obtained in the ordered phase. As mentioned in the paper, "in the infinite-depth limit the mean predictor retains its data-dependence" and is thus not constant (see Appendix D.3 from Xiao et al. [2020]) as it depends on $x$. In addition, the proof that is given relies on the invertibility of $A_{dd}$. Using the notation in Xiao et al. [2020], our setting leads to $p^* = 0$ and $A_{dd} = 0$, the zero matrix. Therefore, the Woodbury identity cannot be applied to conclude the convergence of the mean-predictor. Therein lies the innovation of our proof: $\Theta\^{(L)}\_{\infty} \to 0$ and the limit is not invertible, but the limit of the mean-predictor exists. We thus open the possibility of studying the limit of $\kappa^{(L)}(x^\top X^\top) \left( \kappa^{(L)}\left(X X^\top \right) \right)^{-1}$ as $L \to \infty$ in many architectures. Specifically, this applies to the NTK of an infinitely-wide fully-connected ReLU network, which is beyond the proof of Xiao et al. [2020]. We do not exclude the possibility that a more digestible proof will be found in the future, but we managed to use rough differential equations to circumvent the issue that the limiting kernel is not invertible. To highlight the key insight of our Theorem, we added the mentioned works to the literature review and we emphasize our key contribution by drawing on rough differential equations to solve the non-invertibility issue in limiting kernels. Finally, the list of requirements mentioned at the beginning of section 6 applies to any $\lim_{L \to \infty} \Theta_{\infty}^{(L)}$ that is non-singular for a fully-connected architecture, regardless of the initialization distribution; we only require the ordered phase assumption. We therefore close the gap in the existence of limiting mean-predictors for the ordered phase.

---

> ### Author Response · Authors · 2025-11-24
> **Restricted initialization setting**
>
> In our article, the same initialization as Jacot et al. [2018a] is used, where $W_{ij} \sim \mathcal{N}(0, 1)$ with normalization of layers by $\frac{1}{n_l}$. This is equivalent to initialization $W_{ij} \sim \mathcal{N}(0, \frac{1}{n})$ if each hidden layer has the same width $n$. Theorem 1 and Proposition 3 cited from Jacot et al. [2018a] generalize to other initializations that have $0$ mean and variance $1$. The main insight from our paper concerns the fact that we prove convergence of the mean-predictor as $L \to \infty$ for the limiting kernel of a infinitely-wide fully-connected ReLU network under normal initialization, which is not provided in Xiao et al. [2020] as only the critical phase is studied. As normal initialization is standard practice, we wanted to study the behaviour of such networks as $n, L \to \infty, \frac{L}{n} \to 0$. While we acknowledge that other initializations lead to different limiting kernels for the NTK, the ordered phase described in Xiao et al. [2020] only applies to well-behaved (i.e. invertible) limiting kernels. In the literature review of our paper, we added this reference and clarify the impact of initialization.

---

> ### Author Response · Authors · 2025-11-24
> **Lack of relevant literature discussion**
>
> We added the listed works to the literature review, summarize them, and provide contrast with our results to emphasize the key contributions of our paper.

---

> ### Author Response · Authors · 2025-11-24
> **Misrepresentation of prior results**
>
> The paper by Hanin and Nica [2020] studies the setting $\frac{d}{n} > 0$, where $d$ is depth and $n$ is width. The authors show that $\frac{\mathbb{E} \left[ \mathcal{K}\_{\mathcal{N}}(x, x)\^2 \right]}{\mathbb{E} \left[ \mathcal{K}\_{\mathcal{N}}(x, x)\right]\^2}$, where $\mathcal{K}_{\mathcal{N}}$ is the NTK at initialization, grows exponentially with $\frac{d}{n}$. Specifically, the ratio of the second moment and the first moment squared of the NTK diagonal (i.e. $x = x'$) becomes exponentially large as $n, L$ increase. In our paper, $d = L$ and $n_l$ is  the width of layer $l$. Therefore, the term $\frac{L}{n_l} > 0$ and we can bound it away from $0$ even when $L$ and the widths go to $\infty$. The authors did not specifically take the limit $n, L \to \infty$, and to better reflect the fact that we used their result to characterize this limit, we reformulated the sentence to clear any confusion.

---

> ### Author Response · Authors · 2025-11-24
> **Lack of clarity and technical precision**
>
> ### The initialization distribution is never specified ...
> In the paper, we directly study $\Theta_{\infty}^{(L)}$ from Jacot et al. [2018a] for ReLU activation. This expression implicitly assumes standard normal initialization, which is mentioned at the start of section 4. We agree that readers unfamiliar with the original paper might miss this key assumption. To address this, we added an explicit clarification at the beginning of Section 4.
>
> ### "Case a" (lines 221–222) states that "if data points lie on a unit sphere, ...
> It is true that there is a standard assumption that all points in $X$ are different (on $S^{n_0-1}$), since the kernel $\kappa^{(L)}\left( X X^\top \right)$ needs to be invertible. This is clarified in Section 3. In addition, indeed, Proposition 2 does not exist in Jacot et al. [2018a] but it exists in a subsequent version of the article (Jacot et al. [2018b]). We added the citation to the updated version of the paper. A key assumption for this Proposition is that the kernel is invertible with $L \geq 2$. This is satisfied as we study the range where $L \gg 0$.
>
> ### "Case c" (lines 227–228) refers to `"stereographic ...
> In the paper, we refer to the stereographic projection from a plane to the sphere. The opposite is usually done; to remove any confusion, we instead change the name to "inverse stereographic projection" and also correct $S^{n_0 -1 }$ to $S^{n_0}$, as this was typo. It maps the $n_0$-dimensional plane embedded in a space of one additional dimension ($n_0 + 1$) to the sphere $S^{n_0}$. Mathematically, this is the mapping $\pi_{p} \circ \pi_{e}$, where $\pi_{e}: \mathbb{R}^{n_0} \to \mathbb{R}^{n_0 + 1}, \pi_e(x) = (x_1, \dots, x_{n_0}, 0)$ and $\pi_{p}: \mathbb{R}^{n_0 + 1} \to S^{n_0}$ is the inverse of the stereographic projection from $S^{n_0}$ to $\mathbb{R}^{n_0 + 1}$. We also wish to emphasize that, although not required, this specific embedding on the sphere has the property that two vectors $x, x' \in \mathbb{R}^{n_0}$ are mapped to different points on $S^{n_0}$, while this is not the case for the canonical projection.

---

> > ### Comment · Reviewer_JL34 · 2025-11-26
> >
> > Thank you for your response. Unfortunately, my main concerns remain unaddressed in both the rebuttal and the revision.
> >
> > Regarding the theoretical contribution, the results of Xiao et al. (2020) address exactly the same problem of the infinite-depth limit of the infinite-width NTK. Their assumption that the data-dependent part of the kernel is invertible is essentially the same as the one used in the present paper, since it is satisfied under the same condition that all data points are distinct. The main "message" of this paper (that the mean predictor is well-defined in the ordered phase) appears in Xiao et al. (2020) in lines that go right after Eq. (16). Moreover, Xiao et al. (2020) cover a substantially more general setting, including all initialization phases rather than only the ordered phase. While one may reasonably discuss the level of rigor in Xiao et al. (2020), the present paper does not provide justification that its approach is more rigorous.
> >
> > In addition, the rebuttal and the revision still does not clarify the distinction between *joint* scaling of depth and width (as in Hanin & Nica, 2020) and the *sequential* limits (as in this paper and in Xiao et al., 2020). This distinction is important for understanding the novelty and scope of the results, yet it continues to be presented in an imprecise way.
> >
> > Given the above concerns, I am keeping my score unchanged.

---

> ### Author Response · Authors · 2025-11-27
> **Clarification with the non-invertibility assumption**
>
> Thank you for elaborating on your concerns. In the following, we address these points in detail.
>
> There are two invertibility assumptions in Xiao et al. [2020], while in our work we use only one of them. As the reviewer points out, Xiao et al. [2020] and our work both make the assumption that the limit kernel of the NTK is invertible from the fact that all datapoints are distinct. The other invertibility assumption made by Xiao et al. [2020] requires exploring their proofs as explained in the modified paragraph following the proof of Theorem 3, as well as in the following text. In our work, we use $\Theta\^{(L)}\_{\infty} \left( X X\^\top \right)$, while in Xiao et al. [2020], the authors use the notation $\Theta^{(L)}$  for the same object. However, in the ordered phase, they make another very important and distinct assumption to show the convergence of $P(\Theta)$ (the equivalent of our $\\bar{\Theta}\_{\infty}\^{(L)} \left( X X^\top \right) \left( \Theta_{\infty}^{(L)} \left( X X^\top \right) \right)^{-1}$) to a limiting expression. In Appendix D.3, they show that
> \begin{equation*}
>     \Theta\_{dd}\^{(l)} = p\^* \mathbf{1}\_d \mathbf{1}\_d^\top + l \chi\^l A\_{dd}\^{(l)}
> \end{equation*}
> for some matrix $A\_{dd}\^{(l)}$, where $0 \leq \chi < 1$. They make the very important assumption that $A\_{dd}\^{(l)} \to A_{dd}$ for some invertible matrix $A_{dd}$ to obtain their limiting expression for the mean-predictor. We do not make this invertibility assumption. In fact, in our setting, either 1) $p*=1$ and $A\_{dd} = \mathbf{0}\_{dd}$ for $\lim\_{L \to \infty} \Theta\^{(L)}\_{\infty}$, or 2) $p^*=1$ and $A\_{dd} = \mathbf{0}\_{dd}$ for $\lim_{L \to \infty} \\bar{\Theta}\_{\infty}\^{(L)}$. In both cases, $A_{dd}$ is the all-zero matrix and is not invertible. Therefore, the proof from Xiao et al. [2020] does not apply. It is in fact highly non-trivial to show convergence without this assumption and hence why we believe our work addresses a significant gap in the literature. As for Hanin and Nica [2020], the authors show that the larger the ratio of depth-over-width, the second moment of the NTK's diagonal terms become more significant in comparison to the first moment. In this case, the NTK becomes stochastic and does not converge in probability to the mean as in Jacot et al. [2018a]. We discuss this aspect in the Related Work and we also provide the simple characterization of $\lim_{L \to \infty} \min_{1 \leq l \leq L-1} \frac{L}{n_l}$ as a way to differentiate between the two settings. A finite-width and finite-depth network will approximate the infinite-width, finite-depth network for large widths. By taking $L$ to be as large as we want, as long as we increase the width at an even faster rate, i.e. $L \in o(\min_{1 \leq l \leq L-1})$, we will fall in the setting discussed in our paper; by extension, that would be different from the setting in Hanin and Nica [2020], where the stochasticity becomes relevant as the depth grows faster than the width.

---

### Official Review · Reviewer_bD1j · 2025-10-31

**Soundness:** 3
**Presentation:** 3
**Contribution:** 2
**Rating:** 4
**Confidence:** 4

**Summary:**

This paper analyzes the effect of increasing depth on the Neural Tangent Kernel (NTK) for infinitely wide, overparameterized ReLU networks. While prior work established that wide networks behave like kernel methods, the role of depth remains less understood. The authors prove that as depth increases (with depth growing slower than width), the normalized NTK converges to a matrix of ones, and the kernel regression solution approaches a fixed limit for data on the sphere. They characterize this convergence theoretically using rough path theory and identify key properties enabling generalization to other kernels. Experiments show that while some kernel components converge quickly, the full normalized NTK converges extremely slowly.

**Strengths:**

1. By characterizing the deterministic limit of the NTK and its corresponding solution as depth goes to infinity (under a specific regime), this paper provides novel theoretical insights into the fundamental properties of deep, overparameterized networks, moving beyond the established infinite-width paradigm.

2. The work demonstrates high technical quality through its use of advanced mathematical tools, such as rough path theory, to prove its central convergence theorem (Theorem 3). This sophisticated approach allows the authors to handle the challenging technical obstacle of the kernel matrix becoming singular in the limit, showcasing a rigorous and robust analytical framework.

3. By identifying a list of key properties that lead to the observed limiting behavior, the authors  provide a valuable template for analyzing other kernels and architectures. Furthermore, their empirical evaluation clarifies the practical implications, notably highlighting the extremely slow convergence rate of the normalized NTK, which is a crucial observation for connecting theory to practice.

**Weaknesses:**

1. The empirical validation is minimal, using only synthetic, uniformly distributed data on a sphere for very shallow depths ($L=1$ to $10$). This is insufficient to support the theoretical claim of "extremely slow" convergence, as the chosen depth range is too small to visually demonstrate the asymptotic behavior. To be more compelling, experiments should include benchmarks on real-world datasets and probe much larger depths to empirically quantify the convergence rate and its impact on test performance.

2. While the paper successfully characterizes the existence of a limiting solution, it provides little discussion on what this limit implies for generalization. Does convergence to a fixed limit imply a loss of representation power or a tendency towards simpler functions? A deeper analysis connecting the specific form of the limiting kernel solution to known generalization behaviors would significantly strengthen the significance of the theoretical results.

**Questions:**

1. Your theoretical results establish an "extremely slow" convergence of the normalized NTK to 1, yet your empirical results only show depths up to $L=10$. Could you provide a quantitative estimate of the depth $L$ required for the kernel solution $\kappa_x\kappa^{-1}$ to be within a small $\epsilon$ of its limit on, for instance CIFAR-10? This would greatly help in assessing the practical relevance of your limit for modern deep architectures.

2. Your analysis crucially relies on the depth growing slower than the width to maintain a deterministic NTK. Is this a fundamental requirement for your rough path theory technique to hold, or is it an artifact of the current proof? Could you comment on the feasibility and potential challenges of extending your analysis to the stochastic NTK regime described in Hanin & Nica (2020)?

---

> ### Author Response · Authors · 2025-11-24
> **Reply to weaknesses**
>
> We would like to thank the reviewer for engaging with our paper. We believe addressing these comments helped us to improve the paper by clarifying some important aspects of our results. We also provide new experimental results and these are described in what follows.
>
> ### The empirical validation is minimal ...
> While the experiments do not exhaustively compare various datasets and settings, we use them to show the potential difficulty in experimentally computing the matrix-product $\Theta\_{\infty}\^{(L)} \left( x^\top X^\top \right) \left( \Theta\_{\infty}\^{(L)} \left( X X^\top \right) \right)^{-1}$. The numerical issues that we encountered are the exact reason why we could not compute extremely large depths of say, $10 000$. However, we could reach depths of $100$ and the values on the plots would not change much. Moreover, numerical errors start to become more and more relevant as the matrix $\Theta\^{(L)}\_{\infty}$ gets closer to $0$. In practice, there might exist scenarios where this slow convergence is not an issue, however. Nevertheless, in updated experiments, we use larger values of $L=30$ depicting the very slow convergence over a larger horizon.
>
> ###  While the paper successfully characterizes the existence of a limiting ...
> This limit implies that given 1) a dataset $X$, 2) a particular architecture, and 3) an initialization distribution, we can approximate the output of very large and very deep networks using the limiting expression. This does not imply the loss of representation power: for instance, the targets over the dataset $X$ are perfectly interpolated by this limiting kernel predictor. Since our experiments do not characterize the generalization behaviour directly, we used the limiting solution of a dataset $X$ of size $\left\lvert X \right\rvert = 500$ on MNIST data. We use it to predict the class of digits in the test set and we achieve a high accuracy and F1 score, which we report in the appendix.

---

> ### Author Response · Authors · 2025-11-24
> **Question 1**
>
> By inspection of the proof of Theorem 2, given a small $\delta$, we require a large $L$ so that $\rho^{(L)}$ is close to $1$. The sequence of $\rho^{(L)}$ can be shown to converge to $1$ logarithmically; we added the definition of logarithmic convergence in the appendix. Therefore, it would take a large $L$ to satisfy $\min\left(h(\rho^{(L)}), h'(\rho^{(L)} \right) \geq 1 - \delta$ for a small $\delta$. Similarly, the term $A_{11}^{[L:L+K]} \to 0$ at rate $\approx \frac{1}{K}$. We can conclude that the bound of $\left \lvert \Theta\_{\infty}\^{(L)} \left( X X\^\top \right) - \mathbf{1}\_n \mathbf{1}\_n\^\top \right \rvert$ converges to $0$ at a sublinear rate that is at most as fast as $\frac{1}{K}$. This seems to suggest that the required depth for $\Theta\_{\infty}\^{(L)} \left( X X^\top \right)$ to be close to the all-ones matrix is very large. However, in the proof of Theorem 3, we an observe that the terms $v_{(i,j)}$ converge to $0$ exponentially fast with respect to $\det^{-1} \left( A^{(L)}(t) \right)$. If we get a small determinant, the system in $(5)$ with $u^{(L)}$ (here, we make the dependence of $u$ on $L$ explicit) will be close to its limit $u'(t) = \mathbf{0}\_n$; the solution $u^{(L)}(t)$ would therefore approximate the limiting solution $u_{\infty}(t)$. When we numerically validate the scale of determinants $\det \left( A^{(L)(t) } \right)$, we obtain very small values. We then formulate the hypothesis that with relatively small depths $L$ relative to $\lvert X \rvert = n$, the limiting solution for $\kappa_x \kappa^{-1}$ is well approximated.

---

> ### Author Response · Authors · 2025-11-24
> **Question 2**
>
> If the depth $L \to \infty$ faster than the width, we enter the setting described in Hanin and Nica [2020], where the second moment of the diagonal of the NTK becomes exponentially larger than the first-moment. This translates into a non-constant limiting kernel that does not have a closed-form solution like the one found in Jacot et al. [2018a] as we increase the width (and hence the depth) to infinity. In fact, it becomes impossible to compute a limiting kernel $\Theta\^{(L)}\_{\infty} \left( X X^\top \right)$ as the variance of the diagonal tends to $\infty$. However, in practice, neural networks are to our knowledge almost always wider than they are deeper. For this reason, it became interesting to keep this assumption as both depth and width tend to infinity. We see that the behaviour is very different if the opposite is true.

---

### Official Review · Reviewer_PHxn · 2025-11-01

**Soundness:** 2
**Presentation:** 2
**Contribution:** 2
**Rating:** 4
**Confidence:** 2

**Summary:**

The paper examines the role of network depth through the lens of Neural Tangent Kernel (NTK) theory. Within this framework, this paper shows that the normalized NTK converges to the all-ones matrix as depth increases for data on the sphere (with extensions via projection for more general data), under the assumption that depth grows more slowly than width. Empirical evidence illustrates that this convergence is very slow.

**Strengths:**

The background of NTK theory is clearly presented. Theorems appear to be rigorously argued, and the experiments are conducted to validate them.

**Weaknesses:**

Although the paper considers a different width and depth scaling than Hanin & Nica (2020), I do not think novelty of this result meets the bar for acceptance at this venue. The scope is limited to CNNs, omitting Transformers, now widely used in practice.

The experiments explore depths up to only 10 layers, which is not particularly deep by modern standards.

**Questions:**

If I understand correctly, normalized NTK converge to the all-ones matrix and if so, it would become a deterministic kernel independent of the input data. Is this due to the normalization? And does it imply that the neural network just reduces to a kernel method with a trivial kernel?

Could the authors emphasize the key technical difficulties in the proof of Theorem 3 and explain how they overcame them in plain language?

Missing reference: Please include and discuss “Neural Tangent Kernel Analysis of Deep Narrow Neural Networks” (Lee et al., 2022) in the Related Work section.

**Details Of Ethics Concerns:**

There is no ethics concerns.

---

> ### Author Response · Authors · 2025-11-24
> **Reply to weaknesses**
>
> We would like to thank the reviewer for their comments and thoroughly engaging with our paper. We believe further experiments were needed to better characterize the nature of our theoretical results. We describe these enhancements in details below.
>
> ### Although the paper considers a different width and depth ...
> The objective of this paper is to tackle the difficulty arising in the acquisition of a limiting expression for $\Theta\^{(L)}\_{\infty} \left( x^\top X^\top \right) \left( \Theta\^{(L)}\_{\infty} \left( X X^\top \right) \right)^{-1}$ for a non-invertible $\lim_{L \to \infty} \Theta\_{\infty}\^{(L)}\left(X X^\top \right)$. Different architectures will lead to different kernels $\Theta\^{(L)}\_{\infty}$, but our main contribution opens the possibility to study the aforementioned limiting expression even in degenerate cases. Therefore, we leave to future work the exploration of different architectures and initialization distributions, as they will either satisfy 1) $\lim_{L \to \infty} \Theta\^{(L)}\_{\infty} \left( X X^\top \right)$ invertible, or 2) the criteria of Theorem 3. In case 1, the matrix product that is studied in this article easily admits a limiting expression, and in case 2, the same proof methodology than Theorem 3 is applicable to obtain a limiting expression. When stating our contributions at the end of the Introduction, we mention the use of our proof techniques to obtain the limiting expressions for other architectures or initialization distributions.
>
> ### The experiments explore depths up to only 10 layers ...
> The role of the experiments is to show that reaching such a limiting expression requires very large depths, and this can be observed by the slope of the plot we provided. In addition, there are numerical issues regarding the computation of $\Theta\^{(L)}\_{\infty}\left(x^\top X^\top \right) \left( \Theta\^{(L)}\_{\infty} \left( X X^\top \right) \right)^{-1}$, given that convergence of $\Theta\_{\infty}\^{(L)}$ to $0$ is very fast. This opens the door to the experimental question: "can the computation of this matrix product be done in an efficient way?". Possible avenues of exploration include higher-accuracy floating-point numbers or renormalization at each $L$ during the computation of $\Theta\^{(L)}\_\infty$. This entails an entire new article or package solving this issue and we highly recommend its implementation; we believe this would allow the community to better understand the behaviour of the limiting matrix-product, and thus the output of very large and deep neural networks.

---

> ### Author Response · Authors · 2025-11-24
> **Question 1**
>
> The fact that we get convergence to the all-ones matrix has more to do with the architecture of the network and the activation function that is used (i.e. ReLU). While this seems to imply that the limiting kernel is trivial and data independent, by instead studying the term $\kappa^{(L)}\left( x^\top X \right) \left( \kappa^{(L)} \left( X^\top X \right) \right)^{-1}$, we observe convergence to a limiting expression that depends on $x$. Moreover, this expression perfectly interpolates the labels of dataset $X$ and is non-trivial. In the paper, after Theorem 3, we now stress that the limiting expression of $\kappa^{(L)}\left( x^\top X \right) \left( \kappa^{(L)} \left( X^\top X \right) \right)^{-1}$ depends on $x$.

---

> ### Author Response · Authors · 2025-11-24
> **Question 2**
>
> The main difficulty in the proof of Theorem 3 has to do with the fact that $\\bar{\Theta}\_{\infty}\^{(L)} \to 1$, or similarly that $\Theta\_{\infty}\^{(L)} \to 0$. This prevents the computation of $\lim_{L \to \infty} \Theta\^{(L)}\_{\infty}\left(x^\top X^\top \right) \left( \Theta\_{\infty}\^{(L)}\left( X X^\top \right) \right)^{-1}$, especially using the Woodbury identity. Therefore, we leverage a well-known result in rough differential equations to obtain a limiting expression for the aforementioned matrix product. While $\lim_{L \to \infty} \Theta\^{(L)}\_{\infty} \left( X X^\top \right)$ is not invertible, the matrix product has a limiting expression that depends on $x$. While we focus our proof on the NTK of infinitely wide and deep fully-connected ReLU networks with normal initialization, any other architecture, activation function and initialization distribution combination that falls within the requirements listed at the beginning of section 6 will have such a limiting expression. This is now clearly stated the end of the Introduction when stating our contributions. Moreover, we now explain in plain language the technical difficulties of the proof of Theorem 3 after presenting it.

---

> ### Author Response · Authors · 2025-11-24
> **Question 3**
>
> We added the mentioned article to the Related Work section and we discuss its significance in the present context.

---

> ### Comment · Reviewer_PHxn · 2025-11-26
>
> Thanks the authors for their responses.
>
> The authors state that “The role of the experiments is to show that reaching such a limiting expression requires very large depths, and this can be observed from the slope of the plot we provided.” Could the authors elaborate on why the slope of the plot implies very large depth, and which figure specifically shows this?
>
> In the main text, where do the authors show that ``by instead studying the term  $\kappa^{(L)}\left( x^\top X \right) \left( \kappa^{(L)} \left( X^\top X \right) \right)^{-1}$, authors observe convergence to a limiting expression that depends on $x$''?
>
> The current main text of the PDF is more than 9 pages. Is this allowed?

---

> > ### Author Response · Authors · 2025-11-27
> > **Reply to the official comment**
> >
> > The slope is shown to be logarithmic in the first row, first and second columns of Figure 1. We know from the proof of Theorem 2 that the values of $\\bar{\Theta}\^{(L)}\_{\infty}$ converge to $1$ and we can show this is a logarithmic rate since $h$ and $h'$ converge to $1$ logarithmically. Each additional digit of precision requires larger and larger depths to reach the required level of precision. This is what we wanted to convey by our comment. On the other hand, while this seems to be an issue, the convergence to $\kappa_x \kappa^{-1}$ is shown to be empirically fast, and we hypothesize that it can be shown rigorously in multiple settings.
> >
> > For the second question, we show the result in the paper for $\kappa = \\bar{\Theta}\_{\infty}\^{(L)}$ in Theorem 3. The proof of Theorem 3 states that the limiting expression approaches the value $u\_{\infty}(t)$ that is dependent on $x$. Also, from the statement of Theorem 3, the bound on the norm $\lim_{L \to \infty} \left\lVert \\bar{\Theta}\_{\infty}\^{(L)} \left( x^\top X^\top \right) \left( \\bar{\Theta}\_{\infty}\^{(L)} \left( X X^\top \right) \right)^{-1} \right \lVert_2$  of the limiting expression is dependent on $x$. At the beginning of Section 6, we provide a more general list of assumptions under which the same result will hold.
> >
> > Finally, we believe ICLR allows for an additional page following the reviews as per https://iclr.cc/Conferences/2026/AuthorGuide.

---

> ### Comment · Reviewer_PHxn · 2025-11-27
>
> Thanks the authors for their responses.
>
> However, the experiments only consider depth 30. Could the authors explain why they did not perform experiments with larger depths ?

---

> > ### Author Response · Authors · 2025-11-28
> >
> > We report a depth of $L=30$ to numerically show the convergence to the limiting expression in Figure 1. When we run such experiments up to depth, say $L=200$, we do not obtain more information from the plot. We added a mention of this fact to the the Experiments section. Moreover, for depths larger than $ L=40$, we start to get numerical issues in the inversion of $\kappa^{(L)} \left( X X^\top \right)$, further motivating reporting depths for which the matrix is numerically non-singular (in theory, it is always non-singular). We think that by using higher-precision floating point numbers, it is possible to solve this issue. Otherwise, this could be a research venue to explore in the future. To further cement the fast convergence to the limiting expression and the empirical value of Figure 1, we obtain from the proof of Theorem 3 that a small determinant of $\bar{\Theta}_{\infty}^{(L)} \left( X X^\top \right )$ is sufficient for fast convergence. This is empirically validated on MNIST in Figure 2 in the appendix. We thus hypothesize that a small determinant is obtained for various datasets if they follow the assumptions describing the setting in the paper.

---

### Official Review · Reviewer_UDSf · 2025-11-05

**Soundness:** 2
**Presentation:** 3
**Contribution:** 2
**Rating:** 6
**Confidence:** 3

**Summary:**

This paper provides a theoretical and empirical analysis of how increasing depth affects the Neural Tangent Kernel (NTK) in infinitely wide, fully-connected ReLU networks. The authors show that as depth $L \to \infty$, the normalized limiting NTK $\bar{\Theta}^{(L)}$ converges to a matrix of ones, implying that all inputs become perfectly correlated in the infinite-depth limit. Despite this kernel degeneracy, the closed-form NTK predictor $f(x) = f_0(x) + \Theta_x^{(L)} [\Theta^{(L)}]^{-1} (y - y_0)$ converges to a finite, well-defined limit due to a cancellation effect between the numerator and denominator terms. Theoretical results are supported with toy experiments that visualize convergence rates and verify monotonic correlation growth.
Overall, the paper clarifies the limiting behavior of NTKs with depth and its implications for overparameterized models trained in the kernel regime.

**Strengths:**

-- Rigorous theoretical grounding, with a clean progression from correlation dynamics to kernel limit to predictor stability.

-- Extends NTK theory by isolating the effect of depth, complementing prior focus on width.

--  Elegant use of normalization and rough differential equations to handle singular kernel limits.

-- Empirical plots and numerical validation reinforce theoretical claims (Fig. 1, lines L432--L446).

-- Results implications: very deep, infinitely wide ReLU nets become less expressive, converging to constant mappings.

**Weaknesses:**

-- Experimental section is minimal; only small-scale synthetic tests are shown. A broader empirical sweep would strengthen conclusions.

-- Intuition behind the RDE-based boundedness could be expanded—currently highly technical and somewhat opaque to non-specialists.

-- The convergence rate discussion could quantify how “extremely slow” the approach to 1 is (as noted in L446--L454) using asymptotic bounds.

-- The experimental section and conclusions feel somewhat underdeveloped, leaving the reader wishing for more explicit explanations or additional insights into the implications and experimental findings as per ICLR standards.

**Questions:**

-- Can the authors clarify how the convergence rate of $\phi^{(L)} \to 1$ depends on activation nonlinearity—e.g., would smoother activations yield slower or faster kernel collapse?

-- Since the NTK converges to a constant kernel, does this imply that in the deep infinite-width limit, gradient descent loses any data-dependent inductive bias?

-- Minor comments: 1) L051, L052: Proposition 4 and, Theorem 3 do not have hyperlink (which makes it easy to naviagte) and also does not give much insight as what the actually contributions are by just reading the contributing section, can be written better. 2) same issue with L218, no hyperlinks, poorly written ==>
This, in turn, allow us to aim at characterizing the output of such neural network,as done in the rest of the section. Indeed, from Proposition 2 and Proposition 2 from Jacotet al. (2018), we can immediately observe a few facts regarding the input data:

---

> ### Author Response · Authors · 2025-11-24
> **Reply to weaknesses**
>
> We would like to thank the reviewer for their comments with regards to RDEs and convergence of $\rho^{(L)}$. They were very helpful to identify areas of improvements, specifically the presentation and relevance of theoretical results.
>
> ### Experimental section is minimal ...
> We wish to point out that the synthetic example was meant to illustrate the potential drawbacks on directly analyzing the convergence properties of $\\bar{\Theta}\_{\infty}\^{(L)} \left ( X X^\top \right)$. Even on a real dataset, the convergence will be sublinear in $L$. This slow convergence can nevertheless be disregarded when looking at $\\bar{\Theta}\_{\infty}\^{(L)}\left( x^\top X^\top \right) \left( \\bar{\Theta}\_{\infty}\^{(L)} \left( X X^\top \right) \right)^{-1}$ if the determinant of $\\bar{\Theta}\_{\infty}\^{(L)} \left( X X^\top \right) $ is small. We added experiments with MNIST (multiclass classification for recognizing handwritten digits) and we also perform numerical tests on the convergence of $\det \left( \\bar{\Theta}\_{\infty}\^{(L)} \left( X X^\top \right) \right)$ to empirically show that the limiting expression of Theorem 3 is well approximated at small depth values. As the scope of our paper focuses on the theoretical result from Theorem 3, we stress the importance of further empirical experiments. However, better understanding the convergence duality of $\kappa$ and $\kappa_x \kappa^{-1}$ is beyond the scope of this paper and deserves a comprehensive study.
>
> ### Intuition behind the RDE-based boundedness ...
> We added relevant in the literature review that discuss a similar setting to obtain a limiting expression for $\Theta\_{\infty}\^{(L)}\left( x^\top X^\top \right) \left( \Theta\_{\infty}\^{(L)} \left( X X^\top \right) \right)^{-1}$. For example, Xiao et al. [2020] describe the limit of the ordered phase of the limiting kernel. The authors use the Woodbury identity to obtain a limiting expression for $\Theta\_{\infty}\^{(L)} \left( x^\top X^\top \right) \left( \Theta\_{\infty}\^{(L)} ( X X^\top \right)^{-1}$. Our RDE-based proof circumvents that important issue. We added a description after the proof of Theorem 3 to describe at a high level the idea behind the proof.
>
> ### The convergence rate discussion could ...
> The convergence rate of $\bar{\Theta}\_{\infty}\^{(L)} \left( X X^\top \right)$ to $1$ is sublinear. This is problematic as the proof of Theorem 2 shows that we need very large $L$ values to get a good approximation. However, as shown in our new experiments on the determinant of this matrix, the determinant converges quickly to $0$. From the proof of Theorem 3, equation (5) approximates the limiting solution well, indicating that it is reached at far smaller values of $L$ than would be expected from only looking at the values of $\bar{\Theta}\_{\infty}\^{(L)}$. See sections 6 and 7.
>
> ### The experimental section and conclusions ...
> We refer the reader to the discussion above, regarding new experiments and insights.

---

> ### Author Response · Authors · 2025-11-24
> **Question 1**
>
> We assume the reviewer is referring to $\rho\_{\infty}\^{(L)} \to 1$, as there is no $\phi\^{(L)}$ in the paper. Please let us know if you are referring to some other quantity, and we will gladly go over your question. The convergence implicitly depends on the activation, the initialization distribution ($\mathcal{N}(0, 1)$) and the architecture. Several works out there have studied the value $\\bar{\rho}\^{(L)}$, on which $\\bar{\Theta}\_{\infty}\^{(L)}$ depends, with different activation functions. While this question is hard to answer in a general sense, we can provide a good hand-wavy argument regarding the function $h$ of Lemma 1's proof: if the function $h$ eventually converges to $1$ when its input goes to $1$, but the velocity with which it approaches the value $1$ decreases, then $\\bar{\rho}\^{(L)}$ will approach $1$ slower than in the paper on input $(x, x')$ with $x \neq x'$. In turn, $\\bar{\Theta}\_{\infty}\^{(L)}\left( X^\top X \right)$ will approach $1$ slower on inputs $(x, x')$ with $x \neq x'$; it will still equal $1$ on $(x, x)$. The objective would then be to find the right activation $\sigma$ that slows down this convergence. We can also seek to achieve the opposite: have faster convergence to $1$ through higher velocity of the function $h$ as its input approaches $1$. We note that faster convergence to $1$, or kernel collapse, seems desirable since the term $\mathcal{D}$ in the proof of Theorem 3 will convergence faster to $0$. This results in faster convergence to the limiting solution. It seems that having a smaller Lispchitz constant for the activation $\sigma$ is a productive avenue for achieving faster convergence to $0$. The reasoning behind this hypothesis comes from smaller values of $\\dot{\Sigma}\^{(L)}$ in the product $\\dot{\Sigma}\^{(L)} \Theta\_{\infty}\^{(L)}$ of Theorem 1.

---

> ### Author Response · Authors · 2025-11-24
> **Question 2**
>
> No, the output of the kernel predictor is still data dependent. One immediate fact to observe is that the target over the dataset $X$ are perfectly interpolated. The limiting solution that is derived via Theorem 3 still depends on the input $x$. This serves to highlight the key contribution of our work: while the NTK of an infinitely wide and deep network (under normal initialization) converges to a constant kernel, the output of the kernel predictor is not trivial. In addition, while this sort of result is achieved in Xiao et al. [2020], our proof applies to a more general setting since the limiting kernel falls in the so-called ``ordered phase'' while being non-singular. Hence, the methods of Xiao et al. [2020] cannot be applied and our proof method can better describe what happens in the case of the NTK of infinitely wide and deep fully-connected ReLU networks. We now emphasize this aspect when describing the newly added reference Xiao et al. [2020] in Related Work.

---

> ### Author Response · Authors · 2025-11-24
> **Question 3**
>
> We have added a hyperlinks to Proposition 4 and Theorem 3. We also added a high-level description of what these results offer as a way to better present our contributions after the proof of Theorem 3. We added a hyperlink to Proposition 2 of Jacot et al. [2018a] and rephrased the sentence to make it clearer.

---

### Author Response · Authors · 2025-11-24
**Comments for all reviewers**

We would like to thank each reviewer for their thoughtful comments and questions. The clarifications and enhancements that were required to address these concerns were very useful in improving the state of the paper. In the comments from all reviewers, some points were recurring. We provide a high-level analysis of these questions and our replies. Below, each reviewer may find a personalized response to their individual comment. Changes made to the paper have been highlighted in blue in the new version.

### Prior results for limiting kernels
A recurring comment has to do with the presence in the literature of limiting expressions for both the kernel $\\lim_{L \to \\infty} \kappa^{(L)} \left( X X^\top \right)$ and $\lim_{L \to \infty} \kappa^{(L)}\left( x^\top X^\top \right) \left( \kappa^{(L)} \left( X X^\top \right) \right)^{-1}$. In our setting, the normalized $\\bar{\Theta}\_{\infty}\^{(L)}$ approaches the all-ones matrix. This corresponds to the ``ordered phase'' [Poole et al., 2016]. While this setting corresponds to the usual setting involved in training fully-connected neural networks (especially ReLU networks), to our knowledge, no proof in the literature manages to address the very important fact that $\lim_{L \to \infty} \kappa^{(L)}$ is non-invertible. Therefore, any limiting expression for $\kappa^{(L)} \left( x^\top X^\top \right ) \left( \kappa^{(L)} \left( X X^\top \right) \right)^{-1}$ that is derived using existing methods cannot be applied. We circumvent this issue by employing RDEs to derive a limiting expression, thus closing the gap the in the literature: in the ordered phase, we guarantee the existence of such a limiting expression, regardless of the determinant of $\lim_{L \to \infty} \kappa^{(L)} \left( X X^\top \right)$.

### Initialization
While our setting focuses on a normal initialization of the parameters with layer-wise normalization by $\frac{1}{n_l}$, other initialization schemes exist and can lead to different behaviour for the evolution of the gradient flow solution. These are described in the literature by three regimes: the "ordered phase", the "critical phase", and the "chaotic phase". The setting that most aptly captures the (to our knowledge) typical learning scenario for fully-connected networks remains the ordered phase, as the norm of the NTK remains bounded during initialization (for example, see LeCun initialization). However, as mentioned above, a technical issue arises when studying the limiting expression $\lim_{L \to \infty} \kappa^{(L)}\left( x^\top X^\top \right) \left( \kappa^{(L)} \left( X X^\top \right) \right)^{-1}$ because of the non-invertibility of $\kappa^{(L)}\left( X X^\top \right)$. To our knowledge, no existing proof guarantees the existence of the aforementioned limiting expression for the ordered phase. Our application of RDEs to this problem recovers such a limiting expression regardless of the determinant of $\kappa^{(L)}\left( X X^\top \right)$.

### Experiments and depth
A common mention in the reviews has to do with the lack of experimental results on the depth $L$. Specifically, the lack of real data has been mentioned as weakness. To address these concerns, we have provided further experiments on the MNIST dataset to highlight the similarity of results even on non-synthetic data. We also included numerical experiments on the convergence of both $\\bar{\Theta}\_{\infty}\^{(L)}$ and the limiting solution to Theorem 3.

---

### Meta-Review · Area_Chair_BBWd · 2026-01-05

**Summary:**

The main concerns are:
-  Insufficient experimental validation (only small-scale synthetic datasets, only relatively shallow models considered, missing quantitative convergence analysis  etc.)
- Unclear novelty compared to  Xiao et al. (2020) and related papers.
- Incorrect summary of prior results of Hanin & Nica (2019)
- Lack of clarity and technical precision

**Reviewer Concerns:**

Honestly, I don't think that any of the raised concerns has been addressed in a really convincing way by the authors in their rebuttal. In particular, I think that the more fundamental points of criticism regarding unclear novelty, misinterpretation of related work, etc.  could not be addressed.

**Reviewer Scores:**

Given the many open questions, I don't see many reasons for reviewers to increase their scores substantially.

---

### Decision · Program_Chairs · 2026-01-26

Reject